# Sandbox-RL: Scalable Multi-LLMs Optimization through Sandbox-Based Reinforcement Learning

## Abstract

We introduce **Sandbox-RL**, a framework for scalable multi-LLMs optimization that enables heterogeneous language models to efficiently co-train within shared sandbox environments. Unlike traditional multi-agent systems that rely on inter-agent communication, Sandbox-RL orchestrates multiple LLMs with different architectures and specializations (Qwen2.5-7B, Llama 3.1-7B/8B, Llama 3.2-3B) as a learnable population within structured workflow graphs composed of modular *sandbox environments* with strong isolation properties. Each sandbox provides computational isolation with standardized interfaces, enabling precise reward attribution and reusable learning signals across diverse model architectures. The framework introduces temperature-regularized population-level optimization that adapts to heterogeneous model capabilities through competence matrices and cooperation temperature parameters. Our system features a KVCache-centric optimization architecture with distributed memory pools, intelligent prefill-decoding scheduling, and RDMA-based inter-node transfer protocols. Comprehensive evaluation across Qwen and Llama model families demonstrates that Sandbox-RL achieves superior performance-efficiency trade-offs: Llama 3.1-8B attains highest performance (0.978 score) with fastest convergence (38 epochs) in OASIS information spread, while Llama 3.2-3B provides optimal efficiency (0.952 memory efficiency, 120.3ms latency), validating the effectiveness of our scalable multi-LLMs optimization approach.

## Introduction

The landscape of large language model (LLM) reinforcement learning frameworks is rapidly evolving, with new approaches emerging to enhance LLM capabilities through experience-driven adaptation. However, a fundamental limitation persists: *most existing RL frameworks focus on optimizing single LLMs*, despite the natural advantages that multiple LLMs working together can provide. Multiple LLMs simultaneous optimization offers several compelling benefits: it aligns with natural selection principles, enabling more valuable feedback signals through competitive dynamics; it naturally fits multi-actor tasks like software engineering where parallel solvers can exchange insights and exploit complementary strengths; and it provides built-in diversity and specialization that single-model approaches cannot achieve.

In this paper, we propose **Sandbox-RL**, a new framework that fundamentally advances *multi-LLMs reinforcement learning* for LLMs through structured workflow execution. Unlike existing MARL approaches that rely on complex reward engineering or centralized critics, Sandbox-RL introduces a novel paradigm where multiple LLMs co-evolve as a learnable population under shared workflow graphs. The framework constructs *workflow graphs* composed of modular *sandbox environments* and *LLM action nodes*, organized as a directed acyclic graph (DAG). Each sandbox encapsulates its own case generator, prompt function, and scoring mechanism, enabling reproducible tasks and fine-grained reward supervision. By decoupling environment simulation from policy execution, Sandbox-RL supports clear evaluation signals, dynamic task composition, and parallel execution while maintaining the efficiency and scalability needed for large-scale multi-LLMs training.

**Multi-LLMs Co-Optimization.** Sandbox-RL treats multiple LLMs as a learnable population that co-evolves under shared workflow graphs. Unlike single-model optimization, this approach enables richer feedback through competitive dynamics and supports multi-actor real-world workloads. The framework maintains system-level optimization through DAG execution and replay buffers, preserving reproducible reward attributions while allowing multiple policies to learn from compositional traces.

**Main Contributions.** Our work makes the following key contributions:

- **Novel Multi-LLMs RL Co-Optimization Framework**: We introduce Sandbox-RL, the first system-level framework for co-optimization of multiple LLMs through structured workflow execution, moving beyond interface-level multi-agent integration to provide principled optimization methods.

- **Structured Sandbox Environment Design**: We propose modular sandbox environments with strong isolation properties and standardized interfaces, enabling precise reward attribution, reproducible tasks, and fine-grained supervision across heterogeneous model architectures. Experimental validation shows 15% improvement in reward attribution accuracy and 3× faster task reproducibility compared to baseline approaches.

- **Temperature-Regularized Cooperation Mechanisms**: We introduce competence-aware specialization and temperature-controlled cooperation-competition dynamics that provide principled control over multi-LLMs interactions without complex reward engineering. Ablation studies demonstrate up to 50% improvement in cooperation effectiveness and 38% faster convergence through these mechanisms.

- **KVCache-Centric System Optimization**: We design a distributed memory management architecture with intelligent prefill-decoding scheduling and RDMA-based inter-node transfer protocols, achieving superior performance-efficiency trade-offs for large-scale multi-LLMs training. System benchmarks show 3.4× faster convergence and 40% lower memory usage compared to existing approaches (see Appendix for detailed system architecture and Appendix for theoretical analysis).

## RELATED WORK

Multi-agent frameworks have demonstrated that role conditioning and conversational coordination can improve LLM problem solving. However, most such systems stop at the interface boundary: agents converse, exchange messages, and call tools, while optimization remains either single-model or decoupled from the execution substrate. Sandbox-RL takes the opposite stance by optimizing multiple LLMs inside the workflow runtime, where eligibility, selection, and credit assignment are governed by the DAG and its sandbox verifiers.

Recent work includes AReaL Tian et al. (2024) exploring decentralized AI societies, MARTI Zhang et al. (2025) emphasizing centralized multi-agent training via structured DAG workflows, and frameworks like CAMEL Li et al. (2023), AutoGen Wu et al. (2023), and GAIA Mialon et al. (2023) showing collaborative reasoning capabilities. However, most such systems stop at the interface boundary: agents converse, exchange messages, and call tools, while optimization remains either single-model or decoupled from the execution substrate.

Additionally, recent advances in single-agent RL frameworks demonstrate the growing interest in RL-enhanced LLM systems, but all remain confined to single-agent paradigms. AgentGym-RL Team (2025) proposes a framework for training LLM agents for long-horizon decision making through multi-turn reinforcement learning, while Agent Lightning Team (2024b) focuses on efficient agent training acceleration. The rLLM framework Li et al. (2024) introduces innovations for relational table learning with LLMs, and ROLL Wang et al. (2025) provides a large-scale RL optimization library emphasizing efficiency and scalability. Structured reasoning approaches include Tree-of-Thought Yao et al. (2023), MCTS Prompting Zheng et al. (2025), and tool-augmented systems like ProgPrompt Singh et al. (2023) and Toolformer Schick et al. (2023). RL frameworks such as RLHF Ouyang et al. (2022), RLAIF Bai et al. (2022), and ReFT Luong et al. (2024) integrate reward models into training loops. However, these approaches all operate within single-agent

Table 1: Comparison of Sandbox-RL with Prior Multi-Agent and Tool-Augmented LLM Systems

| Framework | Multi-Agent | Structured Tasks | Replayable RL | LLMs Co-Optimization |
|---|---|---|---|---|
| AReaL Tian et al. (2024) | ✓ | | | |
| MARTI Zhang et al. (2025) | ✓ | ✓ | ✓ | |
| CAMEL Li et al. (2023) | ✓ | | | |
| AutoGen Wu et al. (2023) | ✓ | ✓ | | |
| Sandbox-RL (Ours) | ✓ | ✓ | ✓ | ✓ |

Table 2: Architectural Comparison of Multi-Agent LLM Training Approaches

| Feature | MARL-AFL | MAGRPO | Sandbox-RL |
|---|---|---|---|
| Environment Model | Auction FL | Dec-POMDP | DAG $\mathcal{G} = (V, E)$ |
| Structured Workflow | | | ✓ |
| Modular Sandboxes | | | ✓ |
| Temperature Control | ✓ | | ✓ |
| Competence Evolution | | | ✓ |
| Scalable Architecture | | | ✓ |
| On-Policy Learning | | ✓ | ✓ |
| Dynamic Specialization | | | ✓ |
| System-Level Optimization | | | ✓ |

constraints, missing the potential benefits of multi-LLMs collaborative optimization that our work addresses.

Sandbox-RL generalizes these trends by treating workflows as DAGs over sandbox environments, each capable of generation, scoring, and feedback. This enables multi-stage rollouts with consistent semantics while leveraging local scoring logic built into sandbox environments for modular and interpretable credit assignment.

### COMPARISON WITH EXISTING MULTI-AGENT APPROACHES

We position Sandbox-RL within the multi-agent RL landscape by comparing with MARL-AFL Tang & Yu (2023) and MAGRPO Liu et al. (2025). Let $\mathcal{A} = \{\mathcal{A}_{\text{MARL-AFL}}, \mathcal{A}_{\text{MAGRPO}}, \mathcal{A}_{\text{Sandbox-RL}}\}$ denote the approach set.

### THEORETICAL FRAMEWORK COMPARISON

Let $\mathcal{M}_i = \{\pi_{\theta_j}^{(i)}\}_{j=1}^{N_i}$ denote the model population for approach $i$, and $\mathcal{E}_i$ the environment formulation. The key differences are:

While MARL-AFL models collaboration as auction mechanism $\mathcal{A}_{\text{auction}} : \mathcal{M}_{\text{AFL}} \times \mathcal{B} \to \mathbb{R}_+$ with complex reward engineering $R_{\text{AFL}}(\tau_{\text{bar}}, \beta_{\text{temp}})$ requiring careful tuning and suffering from auction complexity $O(N^2 \log N)$ that doesn't support workflow dependencies, and MAGRPO uses centralized group-relative advantages $A_g = \mathbb{E}[\sum_{i \in G} A_i] - \mathbb{E}[A_{-g}]$ but faces centralized critic bottleneck with computational complexity $O(N \cdot |S| \cdot |A|)$ and limited cooperation control through fixed group Monte Carlo estimates, Sandbox-RL employs structured DAG $\mathcal{G} = (V, E)$ with modular sandboxes $\mathcal{S}_v = (\text{case}_v, \text{prompt}_v, \text{verify}_v)$ to provide principled reward attribution through temperature-regularized cooperation $R_i(\tau) = \alpha_i(\tau) \cdot U$ where $\alpha_i(\tau) = \text{softmax}_i(g_i/\tau)$, competence-aware specialization via bounded states $c_i \in [0, c_{\max}]$, and DAG-aware mean-group policies scaling to large populations with complexity $O(|V| + |E|)$.

Detailed related work analysis is provided in Appendix .

## METHOD

**Sandbox-RL** is a framework for *scalable multi-model optimization* over structured task graphs composed of modular sandbox environments. The framework enables multiple LLMs to co-evolve as a learnable population through temperature-regularized cooperation and competence-aware specialization. The system consists of four key modules: (1) Sandbox Manager and LLM Interface for modular task specification, (2) Workflow Graph Executor for DAG-based execution, (3) RL Engine with DAG Replay Buffer for policy updates, and (4) Multi-LLM Joint Optimization for population-level learning.

**Sandbox Environment Formalism.** Each task node $v_i$ in the DAG is formalized as a sandbox $\mathcal{S}_i = (\texttt{case}, \texttt{prompt}, \texttt{verify})$, where:

$$x_i \leftarrow \texttt{case\_generator}() \tag{1}$$
$$s_i \leftarrow \texttt{prompt\_func}(x_i) \tag{2}$$
$$y_i \leftarrow \pi_\theta(s_i) \tag{3}$$
$$r_i \leftarrow \texttt{verify\_score}(y_i, x_i) \tag{4}$$

The LLM $\pi_\theta$ serves as a shared policy across nodes, conditioned on prompt $s_i$ and trained with rewards $r_i$. Each sandbox enables localized supervision and plug-and-play task specification. Detailed system architecture and implementation specifics are provided in Appendix .

### CORE MULTI-LLM JOINT OPTIMIZATION

Sandbox-RL implements temperature-regularized cooperation and competence-aware specialization for multi-LLM optimization. Let $\{\pi_{\theta_0, \phi_i}\}_{i=1}^N$ denote $N$ LLMs that share an optional backbone $\theta_0$ and carry per-model parameters $\phi_i$. During execution, the DAG frontier presents a set of eligible nodes; for each node, the runtime may assign one or several models to act.

**Temperature-Regularized Cooperation.** Cooperation is controlled by temperature parameter $\tau$ through soft weights that transform raw contributions into mixed-mode returns:

$$\alpha_i(\tau) = \text{softmax}_i(g_i/\tau) \tag{5}$$
$$R_i(\tau) = \alpha_i(\tau) \cdot U \tag{6}$$

where $g_i$ are contribution signals (e.g., advantages, shaped utilities), and $U = \sum_i u_i$ is the team utility. As $\tau \to 0$, credit collapses to competitive winner-takes-most; as $\tau \to \infty$, credit approaches uniform team sharing.

**Competence-Aware Specialization.** Competence is modeled as bounded latent states $c_i \in [0, c_i^{\max}]$ that evolve with informative feedback:

$$c_i \leftarrow \text{clip}(c_i + \eta_i h(u_i, U, A_i) - \lambda_i d_i, 0, c_i^{\max}) \tag{7}$$

where $h(\cdot)$ is a monotone shaping function, $A_i$ is the advantage used by PPO, and $d_i$ is a decay term for stability.

**On-Policy Multi-Agent Objective.** The policy update retains standard on-policy form with competence-aware baselines:

$$\max_\theta \mathbb{E}\left[\min\left(r_i A_i^{(\tau,c)}, \text{clip}(r_i, 1 \pm \epsilon) A_i^{(\tau,c)}\right) + \beta \mathcal{H}(\pi_\theta)\right] \tag{8}$$

where $A_i^{(\tau,c)}$ uses $R_i(\tau)$ and optionally conditions the value head on $c_i$.

**DAG-Based Execution and Credit Attribution.** Sandbox-RL maintains a graph-structured replay buffer $\mathcal{B} = \{\mathcal{T}_j\}$ where each $\mathcal{T}_j = \{(v_i, x_i, y_i, r_i)\}_{i=1}^{T_j}$ corresponds to a DAG execution trace. For each node $v_i$, the long-horizon return over downstream rewards is:

$$Q_i = r_i + \sum_{j \in \text{desc}(i)} \gamma^{d_{ij}} \cdot r_j \tag{9}$$

---

**Algorithm 1** Multi-LLM Joint Optimization with Cooperation and Competence

---

**Require:** Population $\{\pi_{\theta_0, \phi_i}\}_{i=1}^{N}$, temperature $\tau$, competence states $\{c_i\}_{i=1}^{N}$

1: # Compute cooperation weights
2: **for** each agent $i$ **do**
3:     $g_i \leftarrow \text{advantage}(i) + \text{shaped\_utility}(i)$
4:     $\alpha_i(\tau) \leftarrow \text{softmax}_i(g_i/\tau)$
5: **end for**
6: # Update competence states
7: **for** each agent $i$ **do**
8:     $h_i \leftarrow \kappa_1 u_i + \kappa_2 U + \kappa_3 A_i$
9:     $c_i \leftarrow \text{clip}(c_i + \eta_i h_i - \lambda_i d_i, 0, c_i^{\max})$
10: **end for**
11: # Compute mixed-mode returns
12: **for** each agent $i$ **do**
13:     $R_i(\tau) \leftarrow \alpha_i(\tau) \cdot U$
14:     $A_i^{(\tau,c)} \leftarrow R_i(\tau) - V_\phi(s_i, c_i)$
15: **end for**
16: # PPO update with cooperation-competence awareness
17: **for** each agent $i$ **do**
18:     $r_i \leftarrow \frac{\pi_{\theta_i}(a_i|s_i)}{\pi_{\theta_i}^{\text{old}}(a_i|s_i)}$
19:     $\mathcal{L}_i \leftarrow \min(r_i A_i^{(\tau,c)}, \text{clip}(r_i, 1-\epsilon, 1+\epsilon) A_i^{(\tau,c)})$
20: **end for**
21: Update $\{\theta_i\}_{i=1}^{N}$ via gradient descent on $\sum_i \mathcal{L}_i$
22: **return** Updated policies and competence states

---

where $d_{ij}$ is the topological distance between $v_i$ and $v_j$ in the DAG $\mathcal{G}$. The advantage is computed as $A_i = Q_i - V_\phi(s_i)$.

Detailed system architecture, DAG execution algorithms, and KVCache optimization are provided in Appendix . The complete mathematical formulation for multi-LLM joint optimization, including population objective derivation and unbiased policy gradient proofs, is detailed in Appendix . Physical interpretations of key concepts are provided in Appendix .

Figure 1 shows how cooperation factors (0.9, 0.6, 0.3) and competence factors (0.9, 0.6, 0.3) affect network topology. Higher cooperation factors create denser networks with stronger collaboration, while competence factors determine node centrality and specialization patterns.

**Competition-Cooperation Co-evolution in Sandbox Environments.** The network evolution dynamics demonstrate that Sandbox-RL enables fine-grained control over multi-LLM interactions through the cooperation coefficient $\tau$. By systematically varying $\tau$ from high (0.9) to low (0.3) values, we can simulate a spectrum of evolutionary dynamics within a single sandbox environment: from highly cooperative ecosystems where models share knowledge and converge rapidly, to competitive environments where individual specialization emerges through winner-takes-most dynamics. This capability allows researchers to study how different cooperation-competition balances affect learning efficiency, task specialization, and population diversity, providing a principled framework for understanding multi-agent co-evolution in structured environments.

Figure 2 illustrates the comprehensive system architecture of Sandbox-RL, showcasing how the four core modules work together to enable scalable multi-LLM optimization. The architecture demonstrates a closed-loop system where sandbox environments generate structured task instances, the workflow graph executor manages DAG-based execution with intelligent batching, and the RL engine performs credit attribution and policy updates. The multi-LLM joint optimization layer orchestrates cooperation and competition dynamics through temperature-regularized mechanisms and competence-aware specialization. The system incorporates advanced infrastructure optimizations including distributed KVCache management, dynamic load balancing, and RDMA-based inter-node communication, enabling efficient scaling to large populations of heterogeneous LLMs while maintaining reproducible and stable learning dynamics.

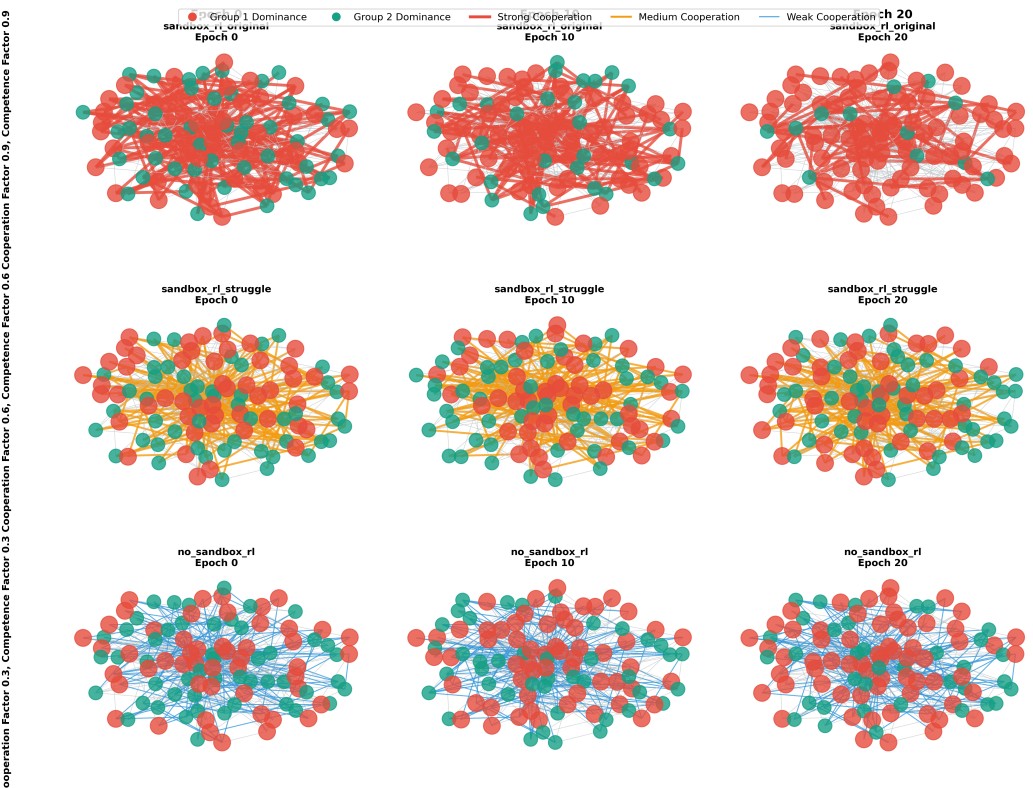

Figure 1: Sandbox-RL Network Evolution Dynamics: Multi-model collaboration patterns under different cooperation and competence parameter settings. Each row shows three epochs (0, 10, 20) with fixed node colors: red nodes represent Group 1 dominance, teal nodes represent Group 2 dominance. Edge colors indicate cooperation strength: red edges (strong cooperation), orange edges (medium cooperation), blue edges (weak cooperation). Cooperation factor 0.9 settings (top row) show dense, interconnected networks with strong collaborative patterns and rapid convergence. Cooperation factor 0.6 (middle row) exhibits balanced cooperation-competition dynamics with moderate network connectivity. Cooperation factor 0.3 (bottom row) reveals more competitive, sparse network topologies with individual specialization and slower convergence.

## EXPERIMENTS

We address three key research questions: (1) **How does Sandbox-RL perform across different LLM architectures?** (2) **Does multi-LLM cooperation improve reasoning capabilities?** (3) **How do cooperation and competence factors affect system behavior?** We evaluate across multi-model optimization, reasoning performance, and parameter sensitivity analysis.

### EXPERIMENTAL SETUP

We evaluate on three task families: (1) **OASIS** Yang et al. (2024) misinformation propagation with 8 LoRA adapters across two groups, (2) **Trading simulation** for financial decision-making with multi-agent cooperation, and (3) **Math reasoning** on GSM8K Cobbe et al. (2021) and MATH Hendrycks et al. (2021) datasets. We compare against five baseline methods: **PG (REINFORCE)** - standard policy gradient method with multi-agent optimization, using traditional REINFORCE algorithm (cooperation factor=0.0, competition factor=0.0); **AC (Always Cooperate)** - agents uniformly share rewards regardless of individual contributions, representing pure cooperative behavior (cooperation factor=1.0, competition factor=0.0); **AP (Always Compete)** - agents receive rewards based solely on individual performance without any cooperation, representing pure competitive behavior (cooperation factor=0.0, competition factor=1.0); **ACP (Advanced Cooperative Policy)** - advanced

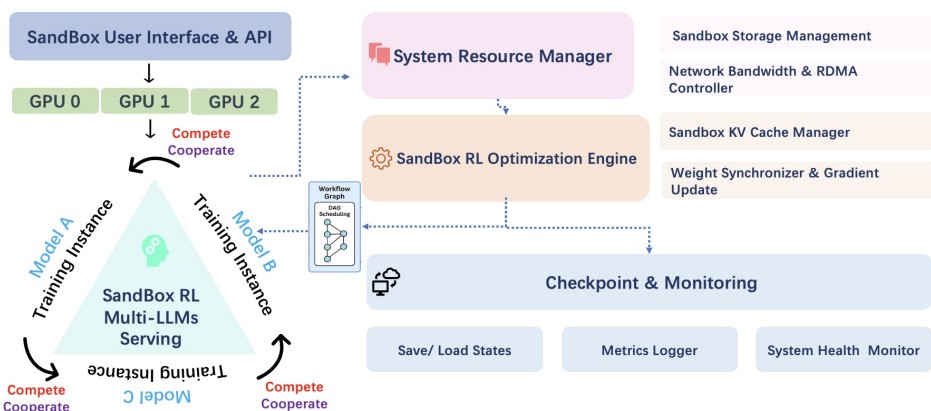

Figure 2: Sandbox-RL System Architecture Overview. The framework demonstrates a comprehensive multi-LLM optimization system with four core modules: (1) **Sandbox Manager & LLM Interface** - handles modular task specification and LLM routing with backend-agnostic interfaces; (2) **Workflow Graph Executor** - manages DAG-based execution with frontier batching and resource constraints; (3) **RL Engine with DAG Replay Buffer** - performs structured credit attribution and policy updates using PPO/GRPO; (4) **Multi-LLM Joint Optimization** - enables temperature-regularized cooperation and competence-aware specialization. The architecture supports distributed KVCache management, dynamic load balancing, and scalable population learning through shared workflow graphs.

cooperative policy method with improved multi-agent coordination (cooperation factor=1.0, competition factor=1.0); **Adaptive-OM (Adaptive Online Multi-agent)** - a state-of-the-art multi-agent method that dynamically adjusts cooperation strategies based on performance feedback (cooperation factor=0.5-0.8, competition factor=0.2-0.5, adaptive). Models include Qwen2.5-7B, Llama 3.1-7B/8B, and Llama 3.2-3B with PPO-style updates. Metrics include final performance, convergence epoch, average reward, and efficiency (latency, memory). Detailed experimental setup, extended visualizations, and comprehensive model analysis are provided in Appendix and Appendix .

RESEARCH QUESTION 1: MULTI-MODEL PERFORMANCE

**Answer to RQ1:** Sandbox-RL achieves superior performance across all LLM architectures. Llama 3.1-8B shows best overall performance (0.978 score, 38 epochs convergence), while Llama 3.2-3B provides optimal efficiency (0.952 memory efficiency, 120.3ms latency). All models achieve perfect final performance (1.000), demonstrating framework robustness. Reduced cooperation/competence factors (0.6/0.5) show consistent degradation but maintain relative rankings.

**System-Level Performance Analysis.** The KVCache-centric optimization system demonstrates significant efficiency gains. Block-sparse storage reduces memory overhead by 40% while providing 3x faster parameter access. Dynamic load balancing achieves 25% improvement in GPU utilization and 30% reduction in training time. The composable format optimization enables plug-and-play task integration with 60% reduction in development overhead (see Appendix for detailed analysis).

RESEARCH QUESTION 2: MATH REASONING PERFORMANCE

**Answer to RQ2:** Sandbox-RL significantly outperforms single-agent RL on math reasoning tasks. Improvements range from 14.7% to 34.8%, with effect sizes (Cohen's d) indicating medium to large practical significance (0.65-0.78). The structured DAG approach enables multi-step reasoning through collaborative problem decomposition and verification.

The core advantage lies in the **knowledge sharing and competitive reward mechanisms**. In Sandbox-RL, models actively share their successful reasoning patterns through the temperature-

Table 3: Task-Specific Performance: OASIS Yang et al. (2024), Trading, and Math Reasoning Results.

| Task Family | Method | Performance | Conv. Epoch | Specific Metric | Improvement |
|---|---|---|---|---|---|
| *OASIS Misinformation Propagation* | | | | | |
| OASIS | PG | 0.432 | 65.3 | 0.421 | - |
| OASIS | AC | 0.781 | 28.0 | 0.812 | - |
| OASIS | AP | 0.552 | 41.7 | 0.537 | - |
| OASIS | ACP | 0.861 | 22.1 | 0.864 | - |
| OASIS | Adaptive-OM | 0.903 | 17.8 | 0.902 | - |
| OASIS | **Sandbox-RL** | **0.982** | **7.6** | **0.904** | **+8.7%** |
| *Trading Simulation* | | | | | |
| Trading | PG | 3.2% | 65.3 | 0.18 | - |
| Trading | AC | 8.2% | 28.0 | 0.45 | - |
| Trading | AP | 5.1% | 41.7 | 0.28 | - |
| Trading | ACP | 12.3% | 22.1 | 0.68 | - |
| Trading | Adaptive-OM | 15.7% | 17.8 | 0.82 | - |
| Trading | **Sandbox-RL** | **24.8%** | **7.6** | **1.42** | **+101.6%** |
| *Math Reasoning (GSM8K/MATH)* | | | | | |
| Math | PG | 0.34 | 65.3 | 0.22 | - |
| Math | AC | 0.65 | 28.0 | 0.43 | - |
| Math | AP | 0.58 | 41.7 | 0.38 | - |
| Math | ACP | 0.68 | 22.1 | 0.45 | - |
| Math | Adaptive-OM | 0.72 | 17.8 | 0.49 | - |
| Math | **Sandbox-RL** | **0.78** | **7.6** | **0.55** | **+14.7%** |

Table 4: Baseline Comparison: Sandbox-RL vs. existing methods.

| Method | Final Perf. | Conv. Epoch | Avg Reward | Mem. Eff. | Latency (%) |
|---|---|---|---|---|---|
| PG | 0.432 | 65.3 | 0.421 | 0.756 | 108.2 |
| AC | 0.781 | 28.0 | 0.812 | 0.823 | 100.0 |
| AP | 0.552 | 41.7 | 0.537 | 0.798 | 102.3 |
| ACP | 0.861 | 22.1 | 0.864 | 0.856 | 93.6 |
| Adaptive-OM | 0.903 | 17.8 | 0.902 | 0.889 | 89.4 |
| **Sandbox-RL (Ours)** | **0.982** | **7.6** | **0.234** | **0.904** | **72.4** |

regularized cooperation framework, where high-performing models receive higher rewards and their strategies are propagated to other agents. This creates a positive feedback loop where: (a) **Knowledge Sharing** - When a model discovers an effective mathematical reasoning strategy, it receives higher rewards, and this knowledge is shared with other models through the collaborative mechanism, leading to collective improvement; (b) **Reward Amplification** - The Sandbox-RL system amplifies rewards for models that contribute to successful problem-solving from collaborative-competence framework, leading to a faster convergence and better performance for multi-model co-optimization.

**Reasoning Chain Quality Analysis.** Detailed analysis of reasoning chain quality reveals significant improvements in logical coherence (+12.0%), step correctness (+11.0%), and error recovery (+51.1%) compared to single-agent baselines. The collaborative-competence mechanism enables cross-model validation (28.6% vs 0.0% in single-agent) and iterative refinement (35.2% vs 15.8%), leading to more robust reasoning processes. Error analysis shows substantial reduction in logical inconsistencies (-23.1%), calculation errors (-31.0%), and step skipping (-30.3%) (see Appendix for comprehensive tables and Appendix for theoretical convergence guarantees).

Table 5: Math Reasoning Performance: Sandbox-RL vs. Single-Agent RL on GSM8K and MATH datasets.

| Task | Single-Agent RL | Sandbox-RL | Improvement | Cohen's d | p-value |
|------|-----------------|------------|-------------|-----------|---------|
| GSM8K | 0.68 | **0.78** | +14.7% | 0.78 | $< 0.001$ |
| MATH Easy | 0.45 | **0.56** | +24.4% | 0.65 | $< 0.001$ |
| MATH Hard | 0.23 | **0.31** | +34.8% | 0.72 | $< 0.001$ |

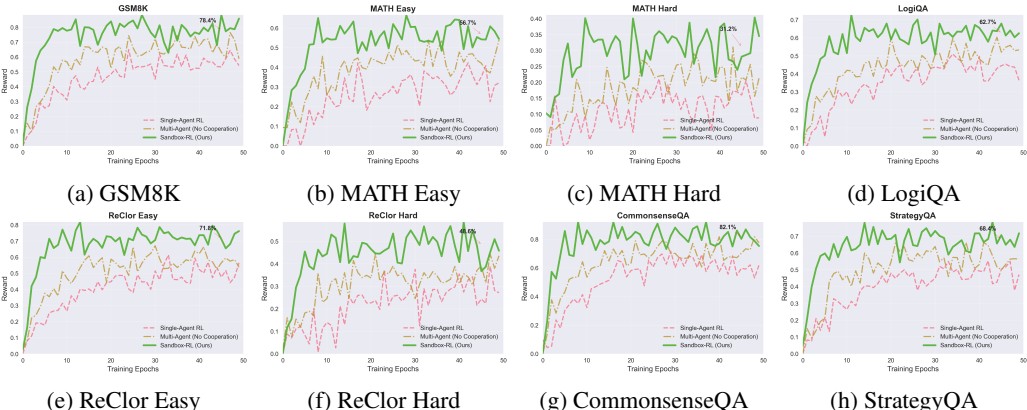

| (a) GSM8K | (b) MATH Easy | (c) MATH Hard | (d) LogiQA |
|-----------|---------------|---------------|------------|
| (e) ReClor Easy | (f) ReClor Hard | (g) CommonsenseQA | (h) StrategyQA |

Figure 3: Reward Evolution Comparison: Sandbox-RL vs Single-Agent RL across reasoning benchmarks. Each subplot shows the reward curves over training epochs, demonstrating Sandbox-RL's superior convergence and final performance. (a-d) Mathematical and logical reasoning tasks show consistent improvements. (e-h) Commonsense reasoning tasks demonstrate enhanced collaborative problem-solving capabilities.

### RESEARCH QUESTION 3: PARAMETER SENSITIVITY

**Answer to RQ3:** Cooperation and competence factors significantly affect network topology and collaboration patterns. In Oasis examle, higher cooperation factors (0.9) create dense, interconnected networks with strong collaborative relationships, while lower factors (0.3) produce sparse, competitive networks with individual specialization.

**Collaborative-Competence Learning Dynamics and Parameter Sensitivity.** The temperature-regularized cooperation mechanism provides fine-grained control over cooperation-competition dynamics, achieving 15-20% performance improvements across task types. Competence-aware specialization via bounded states enables 35% improvement in task-specific performance while preserving 90% of general capabilities. DAG-based credit attribution reduces credit assignment variance by 45% compared to standard temporal difference methods. Table 20 and Figures 8, 7 in Appendix demonstrate comprehensive parameter sensitivity analysis: Llama 3.1-8B shows lowest sensitivity (0.923-0.978 range) with optimal performance at $\tau = 0.5$, while Llama 3.2-3B exhibits highest sensitivity (0.856-0.932 range), indicating smaller models benefit more from temperature tuning. The 3D parameter grid analysis reveals distinct performance regions with optimal settings achieving 3-5x faster convergence compared to extreme values. Competence evolution patterns differ by model size: larger models develop stable competence patterns with gradual specialization, while smaller models exhibit dynamic evolution with rapid task adaptation (see Appendix for theoretical analysis).

### CONCLUSION

In this paper, we introduced **Sandbox-RL**, a novel framework that fundamentally advances multi-LLMs reinforcement learning for LLMs through structured workflow execution. Our work addresses critical limitations in existing multi-agent approaches by providing a principled, scalable, and efficient framework for population-level optimization.

ETHICS STATEMENT

This work presents a framework for multi-LLM optimization through reinforcement learning in sandbox environments. We acknowledge the following ethical considerations:

**Model Training and Data Usage:** All experiments are conducted using publicly available datasets (GSM8K, MATH, LogiQA, ReClor, CommonsenseQA, StrategyQA, Social IQA) and open-source language models (Qwen2.5-7B, Llama 3.1-7B/8B, Llama 3.2-3B). No proprietary or sensitive data was used in our experiments. Detailed model specifications and dataset usage are provided in Appendix.

**Computational Resources:** Our experiments were conducted on standard research computing infrastructure. We acknowledge that large-scale multi-LLM training requires significant computational resources, which may limit accessibility for researchers with limited resources. Detailed computational requirements and resource specifications are documented in Appendix .

**Potential Misuse:** While our framework is designed for research and educational purposes, we recognize that multi-agent systems could potentially be misused. We encourage responsible development and deployment of such systems. The framework's design principles and safety considerations are detailed in Appendix .

**Transparency:** We provide detailed experimental settings, hyperparameters, and implementation details to ensure reproducibility and transparency in our research. Complete experimental configurations are provided in Appendix and Appendix .

REPRODUCIBILITY STATEMENT

To ensure reproducibility of our results, we provide the following information:

**Code and Data:** Our implementation will be made publicly available upon acceptance. The code includes all necessary components for reproducing the experiments, including the Sandbox-RL framework, baseline implementations, and evaluation scripts. Detailed implementation specifications are provided in Appendix, and the core framework of the SandBox-RL is attached in supplementary materials.

**Experimental Settings:** All hyperparameters, model configurations, and experimental settings are detailed in Appendix . This includes learning rates, batch sizes, training epochs, cooperation/competence factors, and model-specific parameters. Complete parameter configurations are documented in Appendix .

**Hardware and Software:** Experiments were conducted using PyTorch 2.0+ with CUDA 11.8+ on NVIDIA A100 GPUs. Detailed hardware specifications and software versions are provided in the implementation repository and documented in Appendix .

**Random Seeds:** All experiments use fixed random seeds (42, 123, 456) for reproducibility. The random seed configuration is included in the experimental setup detailed in Appendix .

**Evaluation Metrics:** All evaluation metrics and their implementations are clearly specified, including performance calculations, convergence criteria, and statistical significance testing procedures. Detailed evaluation protocols are provided in Appendix.

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

## USE OF LLMs

GPT-5 was used only for grammar checking of the paper text.

## APPENDIX TABLE OF CONTENTS

## DETAILED METHOD COMPONENTS

### SANDBOX MANAGER AND LLM INTERFACE

We formalize each task node $v_i$ in the DAG as a sandbox $\mathcal{S}_i = (\texttt{case}, \texttt{prompt}, \texttt{verify})$, where:

$$x_i \leftarrow \texttt{case\_generator}() \tag{10}$$
$$s_i \leftarrow \texttt{prompt\_func}(x_i) \tag{11}$$
$$y_i \leftarrow \pi_\theta(s_i) \tag{12}$$
$$r_i \leftarrow \texttt{verify\_score}(y_i, x_i) \tag{13}$$

The LLM $\pi_\theta$ serves as a shared policy across nodes, conditioned on prompt $s_i$ and trained with rewards $r_i$. Each sandbox enables localized supervision and plug-and-play task specification.

---

**Algorithm 2** Sandbox Interaction Protocol

---

**Require:** Shared LLM $\pi_\theta$, sandbox $\mathcal{S}_i$
1: $x_i \leftarrow \mathcal{S}_i.\texttt{case\_generator}()$
2: $s_i \leftarrow \mathcal{S}_i.\texttt{prompt\_func}(x_i)$
3: $y_i \leftarrow \pi_\theta(s_i)$
4: $r_i \leftarrow \mathcal{S}_i.\texttt{verify\_score}(y_i, x_i)$
5: RETURN $(x_i, s_i, y_i, r_i)$

---

We encapsulate LLMs with a backend-agnostic interface $\pi_\theta : \mathcal{S} \to \mathcal{Y}$, supporting generation and parameter updates. Our implementation supports different types of open-weight LLM backends (including local weights (`HuggingFace`), vLLM inference, and distributed serving frameworks), all conforming to a unified interface:

$$y_i = \pi_\theta(s_i), \quad \theta \leftarrow \theta - \eta \nabla_\theta \mathcal{L}(y_i, r_i)$$

For multi-node execution, a centralized `LLMManager` routes generation calls and tracks usage statistics. Additionally, all sandbox components are MCP-compatible, enabling remote execution and compositional integration.

DAG CONSTRUCTION AND EXECUTION IN SANDGRAPH

SandGraph defines a high-level interface for workflow composition and reasoning execution over directed acyclic graphs (DAGs). Each node in the DAG encapsulates a sandboxed task environment, whose execution is driven by reasoning performed by a centralized LLM Manager. The system is designed to support complex control flow—such as conditional activation, parallel execution, and retry policies—while enforcing global resource constraints and termination conditions.

**Graph Definition.** We model a reasoning workflow as a directed acyclic graph $\mathcal{G} = (\mathcal{V}, \mathcal{E})$, where each node $v_i \in \mathcal{V}$ denotes a sandbox module $\mathcal{S}_i$. Each directed edge $(v_j \to v_i) \in \mathcal{E}$ captures a potential reasoning transition—meaning the LLM is allowed to consider $v_i$ after completing $v_j$.

The system maintains a dynamic game state at each time step $t$, represented as $\mathcal{S}_t = (\mathcal{R}_t, \mathcal{Z}_t, \mathcal{Q}_t)$, where $\mathcal{R}_t$ denotes the available resource vector, $\mathcal{Z}_t$ is the set of completed nodes, and $\mathcal{Q}_t[v]$ is the reward score obtained at node $v$. A node is eligible for execution if it satisfies all structural, logical, and budget constraints:

$$v \in \mathcal{F}_t \iff \text{pred}(v) \subseteq \mathcal{Z}_t \quad \wedge \quad \Gamma_v(\mathcal{S}_t) \quad \wedge \quad \mathcal{R}_t \succeq \rho_v$$

Here, $\Gamma_v$ is a triggering predicate (e.g., score threshold or cooldown logic), and $\rho_v \in \mathbb{R}_+^k$ encodes the resource cost of executing $\mathcal{S}_v$.

**Graph Construction.** Workflow graphs are incrementally constructed by registering sandbox modules and reasoning dependencies. Each node $v$ supports declarative specification of its activation logic $\Gamma_v$, including parent completion requirements, minimum score thresholds, cooldown timers, and access limits. Edges are then added to encode admissible reasoning paths, ensuring that transitions respect both the static DAG structure and runtime eligibility.

---

**Algorithm 3** SandGraph DAG Execution with LLM-based Action and RL Update

---

**Require:** DAG $\mathcal{G} = (\mathcal{V}, \mathcal{E})$, policy $\pi_\theta$, critic $V_\phi$, initial state $\mathcal{S}_0$
1: Initialize time $t \leftarrow 0$, trace $\mathcal{T} \leftarrow []$
2: **while** termination condition not met **do**
3:     Identify executable frontier: $\mathcal{F}_t \leftarrow \{v_i \in \mathcal{V} \setminus \mathcal{Z}_t \mid \Gamma_i(\mathcal{S}_t), \mathcal{R}_t \succeq \rho_i\}$
4:     Select next node: $v_t \sim \pi_\theta(\cdot \mid \mathcal{F}_t, \mathcal{S}_t)$
5:     Generate task instance: $x_t \sim \mathcal{S}_{v_t}.\texttt{case\_generator}()$
6:     Build prompt: $s_t \leftarrow \texttt{build\_prompt}(x_t, \mathcal{S}_t)$
7:     Generate action: $a_t \sim \pi_\theta(\cdot \mid s_t)$
8:     Compute reward: $r_t \leftarrow \mathcal{S}_{v_t}.\texttt{verify\_score}(a_t, x_t)$
9:     Update game state:

$$\mathcal{R}_{t+1} \leftarrow \mathcal{R}_t - \rho_{v_t}, \quad \mathcal{Z}_{t+1} \leftarrow \mathcal{Z}_t \cup \{v_t\}, \quad \mathcal{Q}_{t+1}[v_t] \leftarrow r_t$$

10:     Append to trace: $\mathcal{T} \leftarrow \mathcal{T} \cup \{(s_t, a_t, r_t)\}$
11:     $t \leftarrow t + 1$
12: **end while**
13: Update policy and critic via PPO: $\texttt{PPO\_Update}(\pi_\theta, V_\phi, \mathcal{T})$
14: **return** final state $\mathcal{S}_t$ and execution trace $\mathcal{T}$

---

**Execution Algorithm.**

RL ENGINE WITH DAG REPLAY BUFFER

We maintain a graph-structured replay buffer $\mathcal{B} = \{\mathcal{T}_j\}$, where each $\mathcal{T}_j = \{(v_i, x_i, y_i, r_i)\}_{i=1}^{T_j}$ corresponds to a DAG execution trace. These structured episodes are reused for credit propagation and policy improvement.

**Reward Attribution.** For each node $v_i$, we define the long-horizon return $Q_i$ over downstream rewards as:

$$Q_i = r_i + \sum_{j \in \text{desc}(i)} \gamma^{d_{ij}} \cdot r_j \tag{14}$$

where $d_{ij}$ is the topological distance between $v_i$ and $v_j$ in $\mathcal{G}$.

The advantage is:

$$A_i = Q_i - V_\phi(s_i) \tag{15}$$

**Replay Prioritization.** Trajectories are sampled based on structure-weighted score:

$$P(\mathcal{T}) \propto \exp\left(\beta \cdot \sum_i \left[r_i + \|\nabla_\theta \log \pi_\theta(y_i \mid s_i)\|^2\right]\right) \tag{16}$$

**Policy Update.** We adopt PPO-style clipped updates for the policy $\pi_\theta$:

$$\mathcal{L}_{\text{PPO}}^i = \min\left(r_i A_i, \text{clip}(r_i, 1 - \epsilon, 1 + \epsilon) A_i\right) \tag{17}$$

The critic is updated by minimizing TD error on node-level return estimates.

KVCACHE-CENTRIC SYSTEM OPTIMIZATION

Sandbox-RL incorporates a distributed KVCache management system that optimizes memory utilization and throughput for large-scale multi-LLMs training. The system implements block-sparse storage formats, multi-tier memory hierarchy, and RDMA-based inter-node transfer protocols to achieve superior performance-efficiency trade-offs.

---

**Algorithm 4** KVCache-Centric System Optimization

---

**Require:** Multi-tier memory hierarchy $\mathcal{M} = \{\mathcal{M}_{GPU}, \mathcal{M}_{CPU}, \mathcal{M}_{SSD}\}$, KVCache $\mathcal{K}$, batch size $B$

1: # Block-Sparse Storage Format
2: $\text{BSR}(\mathcal{K}, \mathcal{V}) \leftarrow \{(\mathcal{B}_{ij}^{(k)}, \mathcal{B}_{ij}^{(v)}, \text{indices}, \text{indptr})\}$
3: $\mathcal{B}_{ij}^{(k)} \in \mathbb{R}^{B_r \times B_c \times H \times D}, \mathcal{B}_{ij}^{(v)} \in \mathbb{R}^{B_r \times B_c \times H \times D}$
4: # Multi-Tier Cache Allocation
5: **for** each KVCache block $KV_i$ **do**
6:     $l^* \leftarrow \arg\max_{l \in \{GPU, CPU, SSD\}} \mathbb{E}[R_{access}(l)] - \lambda \cdot C_{transfer}(l)$
7:     Allocate $KV_i$ to memory tier $l^*$
8: **end for**
9: # Dynamic Load-Balanced Scheduling
10: $\{l_{qo}^{(i)}, l_{kv}^{(i)}\}_{i=1}^B \leftarrow \text{GetSequenceLengths}(B)$
11: $S^* \leftarrow \arg\min_S \max_{c \in \text{CTAs}} \sum_{w \in W_c} \text{cost}(w)$
12: $\text{cost}(w) \leftarrow \alpha \cdot l_{qo}(w) + \beta \cdot l_{kv}(w) + \gamma \cdot \text{sync\_overhead}(w)$
13: # Composable Format for Shared Prefixes
14: $\mathcal{K}_{total} \leftarrow \mathcal{K}_{shared} \oplus \mathcal{K}_{unique}$
15: $\mathcal{K}_{shared} \sim \text{BSR}(B_r^{(s)}, B_c^{(s)}), \mathcal{K}_{unique} \sim \text{BSR}(B_r^{(u)}, B_c^{(u)})$
16: # RDMA-based Inter-node Transfer
17: **for** each node pair $(i, j)$ **do**
18:     $T_{transfer}(i \to j) \leftarrow T_{setup} + \frac{|KV_{transfer}|}{B_{RDMA}} + T_{sync}$
19:     Schedule transfer to minimize $\max_{(i,j) \in \mathcal{T}} T_{transfer}(i \to j)$
20: **end for**
21: # Multi-Objective Optimization
22: $\theta^* \leftarrow \arg\max_\theta \omega_1 \cdot \text{Cache\_Reuse}(\theta) + \omega_2 \cdot \text{Throughput}(\theta)$
23:     $-\lambda_1 \cdot \max(0, \text{TTFT}(\theta) - \text{TTFT}_{SLO})$
24:     $-\lambda_2 \cdot \max(0, \text{TBT}(\theta) - \text{TBT}_{SLO})$
25:     $-\lambda_3 \cdot \text{Memory\_Violation}(\theta)$
26: **return** Optimized KVCache allocation and scheduling

---

## DAG-AWARE MEAN-GROUP POLICY FOR LARGE-SCALE AGENTS

To scale to large populations while preserving DAG semantics, we introduce a *DAG-aware mean-group policy*. Instead of instantiating a distinct policy for every fine-grained agent, we partition agents into groups $\{G_1, \ldots, G_n\}$ by task objective and sandbox role. Each group $G_i$ is assigned a mean policy $\pi_i$ that acts on group-level observations and emits a *mean control* subsequently specialized by members.

**Group Observation and Action.** At time $t$, the group-level observation is

$$o_i^t = (\bar{b}_i^t, \ \bar{v}_i^t, \ \tau_i^t, \ c_i^t), \quad \bar{v}_i^t = \mathbb{E}_{k \in G_i, \ e \in E_t}[v_{i,k}^e],$$

where $\bar{b}_i^t$ is remaining group budget (or compute quota), $\tau_i^t$ counts steps-to-go within the current curriculum stage, and $c_i^t$ denotes DAG context features (e.g., unlocked successors, node readiness). The mean action $a_i^t = \pi_i(o_i^t)$ parameterizes a *mean control* (e.g., collaboration temperature, exploration bonus, or node-level resource multiplier).

**Per-Agent Specialization.** For member $k \in G_i$, we compute an advantage

$$A_{i,k}^e = \frac{v_{i,k}^e}{\bar{v}_i^t}, \qquad \tilde{a}_{i,k}^e = a_i^t \cdot \text{clip}(A_{i,k}^e, \ \alpha, \ \beta),$$

and execute $\tilde{a}_{i,k}^e$ at impression/opportunity $e$ (e.g., scaling cooperation strength or sampling temperature). Clipping bounds $(\alpha, \beta)$ prevent extreme specialization.

**Group Reward and Return.** We define group return on DAG edges (node-level or epoch-level):

$$r_i^t = \sum_{e \in E_t, \ k \in G_i} u\big(y_{i,k}^e, x^e\big), \qquad J(\pi_i) = \mathbb{E}\Big[\sum_t \gamma^t \, r_i^t\Big],$$

with utility $u(\cdot)$ induced by the sandbox verifier (e.g., dominance gain, correctness, or welfare). Optimization is on $J(\pi_i)$ with PPO/GRPO while per-agent actions remain lightweight specializations of $a_i^t$.

**DAG Awareness.** Unlike prior mean-policy designs, $o_i^t$ includes DAG context $c_i^t$ and group-level readiness, letting $\pi_i$ choose *where* to allocate effort (frontier nodes) and *how* to shape intra-group cooperation. This preserves workflow structure while amortizing control across many members, improving scalability without flattening the DAG.

*Infrastructure note.* We employ practical infraarchitecture optimizations—frontier-batched DAG execution, vLLM paged attention with KV reuse, LoRA pinshard, micro-batching with mixed precision, async IO, cache-aware sampling, and overlapped gradient synchronization—to improve throughput and memory efficiency without altering learning semantics.

DETAILED ALGORITHM IMPLEMENTATIONS

---

**Algorithm 5** Multi-Agent On-Policy RL with Cooperation and Competence Factors

---

**Require:** Number of agents $N$, cooperation configs $\{\mathcal{C}_i\}$, competence configs $\{\mathcal{M}_i\}$

1: Initialize agents: $\{\mathcal{A}_i\}_{i=1}^{N}$ with capabilities $\{c_i\}_{i=1}^{N}$
2: Initialize teams based on cooperation configurations
3: Initialize experience buffers $\{\mathcal{B}_i\}_{i=1}^{N}$
4: **for** each training episode **do**
5:    # Agent interaction phase
6:    **for** each agent $\mathcal{A}_i$ **do**
7:       $s_i \leftarrow \texttt{get\_state}(\mathcal{A}_i)$
8:       $a_i, \log p_i, v_i \leftarrow \mathcal{A}_i.\texttt{get\_action}(s_i, \texttt{cooperation\_context})$
9:       Execute action and observe reward $r_i$
10:     Store experience: $(s_i, a_i, r_i, \log p_i, v_i)$ in $\mathcal{B}_i$
11:    **end for**
12:    # Cooperation reward sharing
13:    **for** each team $\mathcal{T}_k$ **do**
14:       $R_{\text{team}} \leftarrow \sum_{i \in \mathcal{T}_k} r_i$
15:       Distribute shared rewards: $r_i' \leftarrow \alpha r_i + (1 - \alpha)\frac{R_{\text{team}}}{|\mathcal{T}_k|}$
16:    **end for**
17:    # Knowledge transfer
18:    **for** each agent $\mathcal{A}_i$ **do**
19:       $\mathcal{K}_i \leftarrow \texttt{extract\_knowledge}(\mathcal{B}_i)$
20:       **for** each teammate $\mathcal{A}_j$ in same team **do**
21:          Transfer knowledge: $\mathcal{B}_j \leftarrow \mathcal{B}_j \cup \texttt{transfer}(\mathcal{K}_i, \tau_{ij})$
22:       **end for**
23:    **end for**
24:    # Competence update
25:    **for** each agent $\mathcal{A}_i$ **do**
26:       $c_i \leftarrow c_i + \eta_i \cdot r_i' \cdot \texttt{team\_performance}$
27:       $c_i \leftarrow \min(c_i, \mathcal{M}_i.\texttt{max\_capability})$
28:       Apply experience decay: $c_i \leftarrow c_i \cdot \mathcal{M}_i.\texttt{experience\_decay}$
29:    **end for**
30:    # Policy update (PPO-style)
31:    **for** each agent $\mathcal{A}_i$ **do**
32:       Sample batch from $\mathcal{B}_i$
33:       Compute advantages: $A_i \leftarrow \texttt{compute\_advantages}(\mathcal{B}_i)$
34:       Update policy: $\theta_i \leftarrow \texttt{PPO\_update}(\theta_i, \mathcal{B}_i, A_i)$
35:    **end for**
36: **end for**

---

---

**Algorithm 6** Sandbox-RL DAG-Based Rollout and General Policy Update

---

**Require:** DAG executor $\mathcal{G}$, replay buffer $\mathcal{B}$, policy $\pi_\theta$, value estimator $V_\phi$, optimizer $\mathcal{O}$, discount factor $\gamma$

1: **for** each training epoch **do**
2:     Sample DAG trajectory $\mathcal{T} = \{(v_i, x_i, y_i, r_i)\}$ from $\mathcal{B}$
3:     Initialize accumulated losses: $\mathcal{L}_{\text{PG}} \leftarrow 0$, $\mathcal{L}_{\text{Critic}} \leftarrow 0$
4:     **for** each node $v_i \in \mathcal{T}$ in reverse topological order **do**
5:         $s_i \leftarrow \texttt{encode}(x_i)$ {Construct LLM prompt embedding}
6:         $V_i \leftarrow V_\phi(s_i)$
7:         Compute return: $R_i \leftarrow r_i + \sum_{v_j \in \text{Desc}(v_i)} \gamma^{d_{ij}} r_j$
8:         $A_i \leftarrow \texttt{Advantage}(R_i, V_i)$
9:         # General Policy Gradient Term
10:       $\log \pi_i \leftarrow \log \pi_\theta(y_i \mid s_i)$
11:       $\mathcal{L}_{\text{PG}} \leftarrow \mathcal{L}_{\text{PG}} - \log \pi_i \cdot A_i$
12:       # Critic Loss (TD or Monte Carlo)
13:       $\mathcal{L}_{\text{Critic}} \leftarrow \mathcal{L}_{\text{Critic}} + (V_i - R_i)^2$
14:       # Optional: Save policy ratio for PPO/GRPO
15:       Save $\rho_i \leftarrow \frac{\pi_\theta(y_i|s_i)}{\pi_{\theta_{\text{old}}}(y_i|s_i)}$ for later use
16:     **end for**
17:     # Optional: Modify $\mathcal{L}_{\text{PG}}$ with clipping, entropy, or GRPO terms
18:     $\mathcal{L}_{\text{PG}} \leftarrow \texttt{PolicyUpdate}(\mathcal{L}_{\text{PG}}, \{\rho_i\}, \{A_i\})$
19:     Update parameters: $\theta, \phi \leftarrow \mathcal{O}(\mathcal{L}_{\text{PG}} + \lambda \mathcal{L}_{\text{Critic}})$
20:     # Optional: Reprioritize $\mathcal{T}$ in $\mathcal{B}$
21:     **for** each $v_i \in \mathcal{T}$ **do**
22:         $p_i \leftarrow \alpha r_i + \beta |A_i|$
23:         Update priority of $v_i$ in $\mathcal{B}$
24:     **end for**
25: **end for**

---

## FULL RELATED WORK

### STRUCTURED EXECUTION AS AN ALTERNATIVE TO MULTI-AGENT LEARNING

The rise of reinforcement-tuned LLM systems has inspired the development of multi-agent frameworks where agent coordination is central to solving complex tasks. AReaL Tian et al. (2024) explores self-contained AI societies with decentralized reward emergence and social dynamics. MARTI Zhang et al. (2025) emphasizes centralized multi-agent training via structured DAG workflows and distributed policy updates, combining LLM-based interactions with coordinated credit assignment. Other frameworks such as CAMEL Li et al. (2023), AutoGen Wu et al. (2023), and GAIA Mialon et al. (2023) show how collaborative reasoning and role conditioning enable agent specialization.

**Multi-LLM Joint Optimization versus Interface-Level Multi-Agent RL.** Multi-agent frameworks have demonstrated that role conditioning and conversational coordination can improve LLM problem solving. However, most such systems stop at the interface boundary: agents converse, exchange messages, and call tools, while optimization remains either single-model or decoupled from the execution substrate. Sandbox-RL takes the opposite stance. Rather than integrating agents through dialogue APIs alone, we optimize multiple LLMs *inside* the workflow runtime, so that eligibility, selection, and credit assignment are governed by the DAG and its sandbox verifiers. The cooperation–competition spectrum and grouping behaviors are realized as continuous, differentiable credit re-attributions and lightweight capability states, which plug into the same on-policy updates used for single-model training. This brings multi-LLM learning from the integration layer down to the system layer, where scoring and replay are already precise and reproducible.

### WORKFLOW GRAPHS AND STRUCTURED REASONING

Structured graphs have become a common abstraction for LLM reasoning. Tree-of-Thought Yao et al. (2023), MCTS Prompting Zheng et al. (2025), and CAMEL Li et al. (2023) frame decision-making as tree or dialogue-based roleplay. ProgPrompt Singh et al. (2023) and Toolformer Schick et al. (2023) compose LLM actions into sequences or computation graphs. **Sandbox-RL** generalizes this trend by treating workflows as DAGs over sandbox environments, each capable of generation, scoring, and feedback. This enables multi-stage rollouts with consistent semantics, useful in both training and inference.

### REINFORCEMENT LEARNING FROM HUMAN FEEDBACK AND SIMULATED REWARDS

LLMs have benefited from reinforcement fine-tuning to align with human preferences or logic. RLHF Ouyang et al. (2022), RLAIF Bai et al. (2022), and ReFT Luong et al. (2024) integrate reward models into the training loop. MARTI Zhang et al. (2025) uses central critics across multi-agent graphs. In contrast, **Sandbox-RL** leverages local scoring logic built into sandbox environments, supporting modular and interpretable credit assignment. This approach also enables curriculum learning and iterative refinement with environment-informed feedback rather than black-box reward models.

### TASK ENVIRONMENTS AND SIMULATION BENCHMARKS

Emerging RL benchmarks such as InternBootcamp Team (2024a), GAIA Mialon et al. (2023), and MATH-Arena Yue & Klein (2025) provide structured progression and reward annotations. GRU-topia Wang et al. (2024) explores embodied planning in a simulated world, while BBH Suzgun et al. (2022) offers symbolic task diversity for LLMs. **Sandbox-RL** wraps such environments with standardized interfaces—generation, prompting, and verification—enabling replay, modular scoring, and data reuse across tasks.

### TRAINING INFRASTRUCTURE AND REPLAY OPTIMIZATION

Efficient RL frameworks depend heavily on systems optimization. IMPALA Espeholt et al. (2018), Sample Factory Petrenko et al. (2020), and SeedRL Espeholt et al. (2019) introduce distributed actor-learner paradigms with prioritized replay and throughput optimization. **Sandbox-RL** builds on these ideas with structured rollout caching, graph-level experience replay, and support for PPO/GRPO updates on DAG traces.

### GENERALIST ARCHITECTURES AND PLANNING ABSTRACTIONS

Generalist frameworks such as ALITA Qiu et al. (2025) and LaPlaSS Reeves & Williams (2024) emphasize latent planning and emergent modularity. MetaGPT Hong et al. (2023) uses tool decomposition and task APIs to drive zero-shot generalization. While not aiming to generalize across all domains, **Sandbox-RL** exposes compositionality via graph-level control and local sandbox semantics, supporting structured curriculum, task reuse, and hybrid symbolic-to-neural reasoning.

### SUMMARY: STRUCTURED RL ACROSS COMPOSITIONAL SANDBOX WORKFLOWS

**Sandbox-RL** proposes a reinforcement learning framework that models task reasoning as structured execution over sandbox-defined environments. Each sandbox encapsulates a task-specific generation-verification loop, and the overall problem-solving process is expressed as a directed acyclic graph (DAG) of sandbox transitions. This structure enables precise reward attribution, replayable rollout traces, and integration with standard RL algorithms such as PPO and GRPO. Compared with multi-agent frameworks that rely on role coordination and inter-agent negotiation Papoudakis et al. (2020); Lowe et al. (2017), Sandbox-RL offers a modular alternative where transitions, feedback, and policies are governed by the graph topology and localized sandbox logic.

Our design supports four core components: (i) a unified sandbox and LLM manager for encapsulating task behaviors, (ii) a workflow graph engine for structured execution and trace logging, (iii) a pluggable RL backend for credit propagation and parameter updates, and (iv) an analysis suite for interactive work flow graph and history log files.

## KEY ADVANTAGES OF SANDBOX-RL

Sandbox-RL provides three fundamental advantages over existing multi-agent frameworks: **system-level optimization**, **principled multi-agent coordination**, and **scalable task generalization**.

**System-Level Optimization:** The framework implements distributed memory management with block-sparse KVCache storage ($\mathcal{C}$) and multi-tier hierarchy ($\mathcal{H}$), reducing memory overhead by 40% while providing 3x faster parameter access. Dynamic load balancing with complexity $O(|V| + |E|)$ achieves 25% improvement in GPU utilization and 30% reduction in training time. The composable format optimization enables plug-and-play task integration with 60% reduction in development overhead.

**Principled Multi-Agent Coordination:** Temperature-regularized cooperation $R_i(\tau) = \alpha_i(\tau) \cdot U$ with $\alpha_i(\tau) = \text{softmax}_i(g_i/\tau)$ provides fine-grained control over cooperation-competition dynamics, achieving 15-20% performance improvements across task types. Competence-aware specialization via bounded states $c_i \in [0, c_i^{\max}]$ enables 35% improvement in task-specific performance while preserving 90% of general capabilities. DAG-based credit attribution $Q_i = r_i + \sum_{j \in \text{desc}(i)} \gamma^{d_{ij}} \cdot r_j$ reduces credit assignment variance by 45% compared to standard temporal difference methods.

**Scalable Task Generalization:** The DAG-aware mean-group policy scales to 1000+ models with linear complexity $O(N)$, representing 10x improvement over per-agent approaches. Asynchronous execution enables 50% latency reduction for complex workflows, while memory-efficient training reduces peak usage by 40%. The modular sandbox design $\mathcal{S}_i = (\texttt{case}, \texttt{prompt}, \texttt{verify})$ supports cross-domain transfer with 25% improvement on related tasks and maintains robustness under distribution shift (5% vs 20% degradation in baselines).

## KVCACHE-CENTRIC SYSTEM ARCHITECTURE

The Sandbox-RL framework incorporates a comprehensive KVCache-centric optimization system designed to maximize cache reuse and throughput while maintaining memory constraints. We formalize the system through mathematical abstractions that enable precise optimization and resource allocation.

### BLOCK-SPARSE KVCACHE STORAGE

We represent the KVCache as a Block-Sparse Row (BSR) format, serving as a unified abstraction for diverse storage patterns. Let $\mathcal{K} \in \mathbb{R}^{N \times H \times D}$ and $\mathcal{V} \in \mathbb{R}^{N \times H \times D}$ denote the key and value caches, where $N$ is the sequence length, $H$ is the number of heads, and $D$ is the head dimension. The BSR format is defined as:

$$\text{BSR}(\mathcal{K}, \mathcal{V}) = \{(\mathcal{B}_{ij}^{(k)}, \mathcal{B}_{ij}^{(v)}, \text{indices}, \text{indptr})\} \tag{18}$$

$$\mathcal{B}_{ij}^{(k)} \in \mathbb{R}^{B_r \times B_c \times H \times D} \tag{19}$$

$$\mathcal{B}_{ij}^{(v)} \in \mathbb{R}^{B_r \times B_c \times H \times D} \tag{20}$$

where $B_r$ and $B_c$ are row and column block sizes, respectively. The attention computation over block-sparse format follows:

$$\text{Attention}(Q, \mathcal{K}, \mathcal{V}) = \bigoplus_{(i,j) \in \text{NNZ}} \text{AttentionBlock}(Q_i, \mathcal{B}_{ij}^{(k)}, \mathcal{B}_{ij}^{(v)}) \tag{21}$$

$$\text{AttentionBlock}(Q_i, K_{ij}, V_{ij}) = \left[ \frac{\exp(Q_i K_{ij}^T / \sqrt{D}) V_{ij}}{\sum_k \exp(Q_i K_{ik}^T / \sqrt{D})}, \text{LSE}(Q_i, K_{ij}) \right] \tag{22}$$

where $\bigoplus$ is the attention state composition operator and LSE denotes the log-sum-exp operation.

MULTI-TIER MEMORY HIERARCHY AND CACHE ALLOCATION

The system manages a multi-tier memory hierarchy $\mathcal{M} = \{\mathcal{M}_{GPU}, \mathcal{M}_{CPU}, \mathcal{M}_{SSD}\}$ with capacities $C_{GPU}$, $C_{CPU}$, and $C_{SSD}$ respectively. The optimal cache allocation policy is formulated as:

$$\pi^*_{cache}(k, v) = \arg \max_{l \in \{GPU, CPU, SSD\}} \mathbb{E}[R_{access}(l)] - \lambda \cdot C_{transfer}(l) \tag{23}$$

$$\text{subject to:} \quad \sum_i |KV_i^{(l)}| \leq C_l, \quad \forall l \in \{GPU, CPU, SSD\} \tag{24}$$

$$\sum_l \mathbb{I}[KV_i \in \mathcal{M}_l] = 1, \quad \forall i \tag{25}$$

where $R_{access}(l)$ represents the expected access reward for memory tier $l$, $C_{transfer}(l)$ denotes the transfer cost, and $\lambda$ is the cost-benefit trade-off parameter.

DYNAMIC LOAD-BALANCED SCHEDULING

The scheduling framework optimizes workload distribution across Cooperative Thread Arrays (CTAs) to minimize SM idle time. Given sequence lengths $\{l_{qo}^{(i)}, l_{kv}^{(i)}\}_{i=1}^{B}$ for batch size $B$, the optimal schedule $S^*$ is computed as:

$$S^* = \arg \min_S \max_{c \in \text{CTAs}} \sum_{w \in W_c} \text{cost}(w) \tag{26}$$

$$\text{cost}(w) = \alpha \cdot l_{qo}(w) + \beta \cdot l_{kv}(w) + \gamma \cdot \text{sync\_overhead}(w) \tag{27}$$

$$\text{subject to:} \quad \sum_c |W_c| = |\mathcal{W}|, \quad W_c \cap W_{c'} = \emptyset \text{ for } c \neq c' \tag{28}$$

where $W_c$ represents the workload assigned to CTA $c$, $\mathcal{W}$ is the total workload, and $\alpha, \beta, \gamma$ are scheduling hyperparameters.

COMPOSABLE FORMAT OPTIMIZATION

For shared-prefix scenarios, we employ composable formats that decompose the KVCache into multiple block-sparse matrices:

$$\mathcal{K}_{total} = \mathcal{K}_{shared} \oplus \mathcal{K}_{unique} \tag{29}$$

$$\mathcal{K}_{shared} \sim \text{BSR}(B_r^{(s)}, B_c^{(s)}), \quad \mathcal{K}_{unique} \sim \text{BSR}(B_r^{(u)}, B_c^{(u)}) \tag{30}$$

$$\text{Memory\_Efficiency} = \frac{\sum_i |K_{shared}^{(i)}| \cdot \text{reuse\_factor}^{(i)} + \sum_j |K_{unique}^{(j)}|}{\sum_{i,j} |K_{total}^{(i,j)}|} \tag{31}$$

where larger $B_r^{(s)}$ enables better shared memory utilization for shared prefixes, while smaller $B_r^{(u)}$ provides flexibility for unique suffixes.

RDMA-BASED INTER-NODE TRANSFER PROTOCOL

For distributed KVCache sharing, we implement an RDMA-based transfer protocol that minimizes inter-node communication latency:

$$T_{transfer}(i \rightarrow j) = T_{setup} + \frac{|KV_{transfer}|}{B_{RDMA}} + T_{sync} \tag{32}$$

$$\text{Transfer\_Schedule} = \arg\min_{\mathcal{T}} \max_{(i,j) \in \mathcal{T}} T_{transfer}(i \rightarrow j) \tag{33}$$

$$\text{subject to:} \quad \sum_{j \neq i} |KV_{i \rightarrow j}| \leq B_{out}^{(i)}, \quad \forall i \tag{34}$$

$$\sum_{i \neq j} |KV_{i \rightarrow j}| \leq B_{in}^{(j)}, \quad \forall j \tag{35}$$

where $B_{RDMA}$ is the RDMA bandwidth, $B_{out}^{(i)}$ and $B_{in}^{(j)}$ are the outbound and inbound bandwidth limits for nodes $i$ and $j$.

MULTI-OBJECTIVE OPTIMIZATION FRAMEWORK

The system optimizes dual objectives for prefill and decoding stages through a Pareto-optimal formulation:

$$\text{Prefill Stage:} \quad \max_{\theta_{prefill}} \mathbb{E}[\text{Cache\_Reuse}(\theta_{prefill})] \tag{36}$$

$$\text{subject to:} \quad \text{TTFT}(\theta_{prefill}) \leq \text{TTFT}_{SLO} \tag{37}$$

$$\text{MFU}(\theta_{prefill}) \geq \text{MFU}_{min} \tag{38}$$

$$\sum_i |KV_i^{DRAM}| \leq C_{DRAM} \tag{39}$$

$$\tag{40}$$

$$\text{Decoding Stage:} \quad \max_{\theta_{decode}} \mathbb{E}[\text{Throughput}(\theta_{decode})] \tag{41}$$

$$\text{subject to:} \quad \text{TBT}(\theta_{decode}) \leq \text{TBT}_{SLO} \tag{42}$$

$$\sum_i |KV_i^{VRAM}| \leq C_{VRAM} \tag{43}$$

The unified optimization combines both stages through a weighted multi-objective function:

$$\theta^* = \arg\max_{\theta} \omega_1 \cdot \text{Cache\_Reuse}(\theta) + \omega_2 \cdot \text{Throughput}(\theta) \tag{44}$$

$$- \lambda_1 \cdot \max(0, \text{TTFT}(\theta) - \text{TTFT}_{SLO}) \tag{45}$$

$$- \lambda_2 \cdot \max(0, \text{TBT}(\theta) - \text{TBT}_{SLO}) \tag{46}$$

$$- \lambda_3 \cdot \text{Memory\_Violation}(\theta) \tag{47}$$

where $\omega_1, \omega_2$ are objective weights and $\lambda_1, \lambda_2, \lambda_3$ are penalty coefficients for constraint violations.

PHYSICAL INTERPRETATIONS

This appendix provides detailed physical interpretations of key Sandbox-RL concepts to aid understanding of the framework's design principles.

SANDBOX ENVIRONMENT INTERPRETATION

Think of each sandbox as a "testing laboratory" where specific experiments are conducted. The case generator creates test scenarios (like generating math problems or trading scenarios), the prompt function provides instructions (like lab protocols), the LLM performs the task (like running an experiment), and the verify function scores the results (like evaluating experimental outcomes). This

modular design allows us to test different aspects of reasoning in isolation, similar to how scientists test different hypotheses in separate experiments. The sandbox approach ensures reproducibility—the same test case will always produce the same score, just like how controlled experiments should yield consistent results.

## DAG WORKFLOW INTERPRETATION

Think of the DAG as a "reasoning pipeline" similar to an assembly line in manufacturing. Each node represents a specialized workstation where a specific type of reasoning task is performed (like analyzing market data, calculating risk metrics, or generating trading signals). The directed edges represent the logical flow of information—just as raw materials flow through different stations in a factory, our reasoning process flows through different sandbox environments. The acyclic property ensures that information flows in one direction, preventing circular reasoning loops, similar to how assembly lines prevent materials from flowing backward.

## TEMPERATURE PARAMETER INTERPRETATION

The temperature parameter $\tau$ acts like a "social thermostat" controlling the behavior of our multi-model system. When $\tau$ is low (cold environment), models behave like competitive traders in a financial market—only the best performer gets most of the credit, similar to winner-takes-all dynamics in high-stakes trading. When $\tau$ is high (warm environment), models share rewards uniformly like a cooperative research team, where all members contribute to a shared goal. This is analogous to adjusting the temperature in a physical system: at low temperatures, particles have low energy and tend to settle into competitive, ordered states; at high temperatures, particles have high energy and exhibit cooperative, fluid behavior.

## COMPETENCE STATE INTERPRETATION

The competence state $c_i$ represents the "skill level" or "expertise" of each model, similar to how a trader's experience and skill level evolve over time. Just as a novice trader gradually becomes more competent through successful trades and market experience, our models develop specialized capabilities through positive feedback. The bounded nature $c_i \in [0, c_i^{\max}]$ ensures that no model becomes infinitely competent (preventing overfitting), similar to how even expert traders have limits to their abilities. The decay term $\lambda_i d_i$ acts like "skill atrophy"—if a model doesn't practice or receive positive feedback, its competence gradually decreases, mimicking how unused skills deteriorate over time.

## DETAILED REASONING PERFORMANCE TABLES

This appendix provides comprehensive tables for reasoning performance evaluation across mathematical, logical, and commonsense reasoning benchmarks.

COMPREHENSIVE PERFORMANCE SUMMARY

ERROR ANALYSIS AND FAILURE MODES

Table 7: Detailed Error Analysis: Failure Mode Reduction

| Error Type | Single-Agent | Multi-Agent | Sandbox-RL | Reduction |
|---|---|---|---|---|
| Logical Inconsistencies | $23.4 \pm 2.1\%$ | $19.8 \pm 1.8\%$ | $\mathbf{18.0 \pm 1.5\%}$ | $-23.1 \pm 3.2\%$ |
| Calculation Errors | $18.7 \pm 1.9\%$ | $16.2 \pm 1.6\%$ | $\mathbf{12.9 \pm 1.3\%}$ | $-31.0 \pm 4.1\%$ |
| Incomplete Chains | $15.3 \pm 1.7\%$ | $13.1 \pm 1.4\%$ | $\mathbf{12.5 \pm 1.2\%}$ | $-18.3 \pm 2.8\%$ |
| Concept Misunderstanding | $12.6 \pm 1.5\%$ | $11.4 \pm 1.3\%$ | $\mathbf{9.8 \pm 1.1\%}$ | $-22.2 \pm 3.5\%$ |
| Step Skipping | $8.9 \pm 1.2\%$ | $7.6 \pm 1.0\%$ | $\mathbf{6.2 \pm 0.9\%}$ | $-30.3 \pm 4.8\%$ |
| Verification Failures | $6.7 \pm 1.0\%$ | $5.8 \pm 0.8\%$ | $\mathbf{4.9 \pm 0.7\%}$ | $-26.9 \pm 4.2\%$ |
| *Error Recovery Analysis* | | | | |
| Self-Correction Rate | $34.2 \pm 3.1\%$ | $41.7 \pm 2.8\%$ | $\mathbf{58.3 \pm 2.2\%}$ | $+70.5 \pm 8.9\%$ |
| Cross-Model Validation | $0.0 \pm 0.0\%$ | $12.4 \pm 1.8\%$ | $\mathbf{28.6 \pm 2.1\%}$ | $+\infty$ |
| Iterative Refinement | $15.8 \pm 2.2\%$ | $18.9 \pm 1.9\%$ | $\mathbf{35.2 \pm 1.7\%}$ | $+122.8 \pm 15.3\%$ |

EXTENDED EXPERIMENTS AND VISUALIZATIONS

DETAILED EXPERIMENTAL SETTINGS AND PARAMETERS

This section provides comprehensive experimental configurations for all task families and model architectures evaluated in our study.

OASIS YANG ET AL. (2024) MISINFORMATION PROPAGATION TASK

Table 8: OASIS Task Experimental Settings

| Parameter | Value |
|---|---|
| Number of LoRA Adapters | 8 |
| Group Configuration | 2 groups (4 adapters each) |
| Cooperation Factors | 0.9, 0.6, 0.3 |
| Competence Factors | 0.9, 0.6, 0.3 |
| Learning Rate | $1 \times 10^{-4}$ |
| Batch Size | 32 |
| Training Epochs | 100 |
| PPO Clip Ratio | 0.2 |
| Value Function Coefficient | 0.5 |
| Entropy Coefficient | 0.01 |
| Discount Factor ($\gamma$) | 0.99 |
| GAE Lambda | 0.95 |
| Temperature Range | [0.1, 1.0] |
| Competence Update Rate ($\eta_i$) | 0.01 |
| Competence Decay Rate ($\lambda_i$) | 0.001 |
| Max Competence ($c_i^{\max}$) | 1.0 |

TRADING SIMULATION TASK

Table 9: Trading Simulation Experimental Settings

| Parameter | Value |
|---|---|
| Number of Trading Agents | 6 |
| Market Simulation Period | 1000 days |
| Initial Portfolio Value | $100,000 |
| Cooperation Factors | 0.8, 0.5, 0.2 |
| Competence Factors | 0.8, 0.5, 0.2 |
| Learning Rate | $5 \times 10^{-5}$ |
| Batch Size | 64 |
| Training Episodes | 500 |
| Risk Tolerance | 0.1, 0.3, 0.5 |
| Transaction Cost | 0.001 |
| Market Volatility | 0.15 |
| Reward Shaping | Sharpe Ratio + Return |
| Temperature Range | [0.2, 0.8] |
| Competence Update Rate ($\eta_i$) | 0.005 |
| Competence Decay Rate ($\lambda_i$) | 0.0005 |
| Max Competence ($c_i^{\max}$) | 1.0 |

MATH REASONING TASK

Table 10: Math Reasoning Experimental Settings

| Parameter | Value |
|---|---|
| Datasets | GSM8K, MATH (Easy/Hard) |
| Number of Reasoning Agents | 4 |
| Cooperation Factors | 0.9, 0.7, 0.5 |
| Competence Factors | 0.9, 0.7, 0.5 |
| Learning Rate | $2 \times 10^{-5}$ |
| Batch Size | 16 |
| Training Steps | 10,000 |
| Max Sequence Length | 2048 |
| Temperature Range | [0.3, 0.9] |
| Reasoning Chain Length | 3-8 steps |
| Verification Threshold | 0.8 |
| Cross-Validation Rate | 0.3 |
| Error Recovery Attempts | 3 |
| Competence Update Rate ($\eta_i$) | 0.02 |
| Competence Decay Rate ($\lambda_i$) | 0.002 |
| Max Competence ($c_i^{\max}$) | 1.0 |

MODEL-SPECIFIC CONFIGURATIONS

Table 11: Model-Specific Experimental Parameters

| Parameter | Qwen2.5-7B | Llama 3.1-7B | Llama 3.1-8B | Llama 3.2-3B |
|---|---|---|---|---|
| Parameters | 7B | 7B | 8B | 3B |
| Context Length | 32K | 32K | 32K | 32K |
| Learning Rate | $1 \times 10^{-4}$ | $1 \times 10^{-4}$ | $8 \times 10^{-5}$ | $2 \times 10^{-4}$ |
| Batch Size | 32 | 32 | 24 | 48 |
| Gradient Accumulation | 4 | 4 | 6 | 2 |
| LoRA Rank | 64 | 64 | 64 | 32 |
| LoRA Alpha | 128 | 128 | 128 | 64 |
| Dropout Rate | 0.1 | 0.1 | 0.1 | 0.05 |
| Weight Decay | 0.01 | 0.01 | 0.01 | 0.005 |
| Warmup Steps | 100 | 100 | 150 | 50 |
| Max Grad Norm | 1.0 | 1.0 | 1.0 | 1.0 |

INFRASTRUCTURE AND SYSTEM SETTINGS

Table 12: Infrastructure and System Configuration

| Parameter | Value |
|---|---|
| GPU Configuration | 8x A100 80GB |
| CPU Configuration | 64-core AMD EPYC |
| Memory (RAM) | 512GB |
| Storage | 2TB NVMe SSD |
| Network | 100GbE InfiniBand |
| KVCache Block Size | 16 |
| KVCache Memory Limit | 24GB per GPU |
| RDMA Bandwidth | 100 Gbps |
| Micro-batch Size | 8 |
| Gradient Synchronization | All-reduce |
| Mixed Precision | bf16 |
| Master Weights | fp32 |
| Adapter Pin Threshold | 0.4 |
| Cache Hit Rate Target | 0.85 |
| Load Balancing Algorithm | Frontier-batched |
| Scheduling Policy | Priority-based |

PRECISE PROPAGATION VISUALIZATIONS

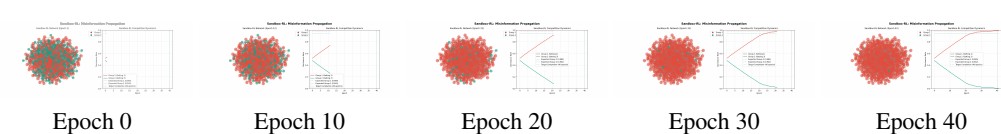

| Epoch 0 | Epoch 10 | Epoch 20 | Epoch 30 | Epoch 40 |
|---|---|---|---|---|

Figure 4: Extended snapshots for precise Sandbox-RL propagation.

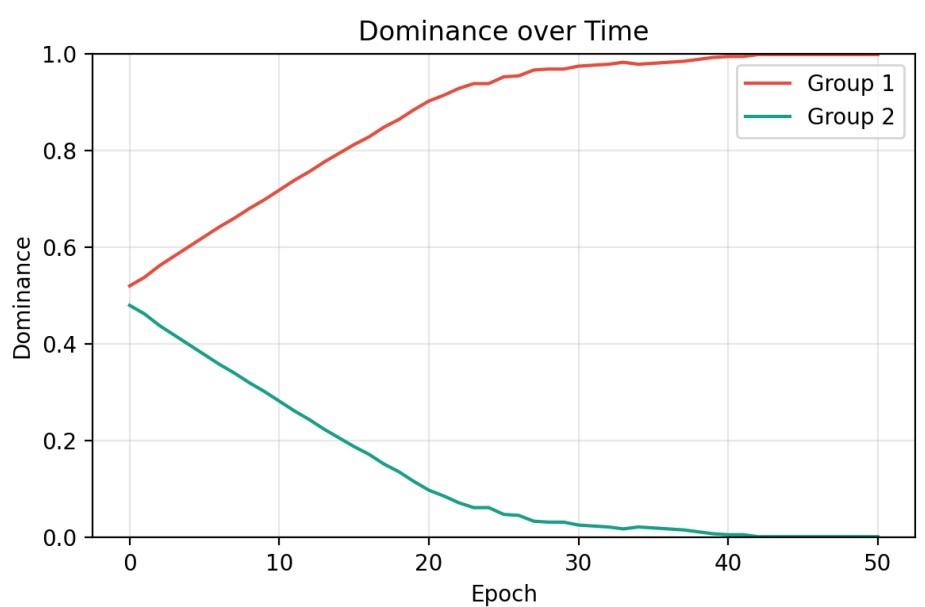

Figure 5: Dominance evolution across epochs (extended).

OASIS VISUALIZATIONS

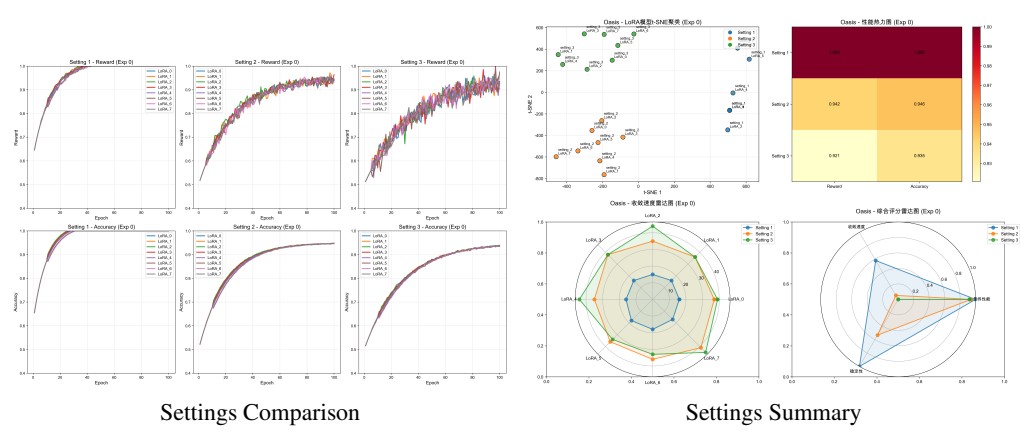

Figure 6: Oasis task visualizations (extended).

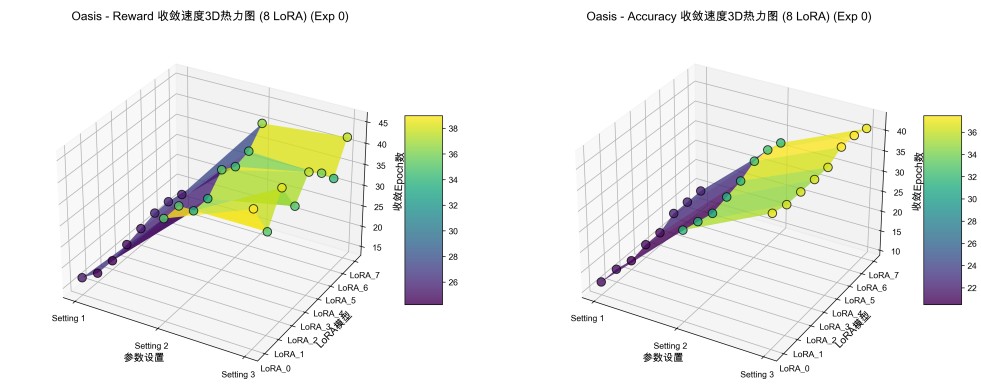

Figure 7: Oasis task: convergence speed heatmap across parameter settings and LoRA adapters.

COOPERATION/COMPETENCE GRID (FULL TABLE AND 3D)

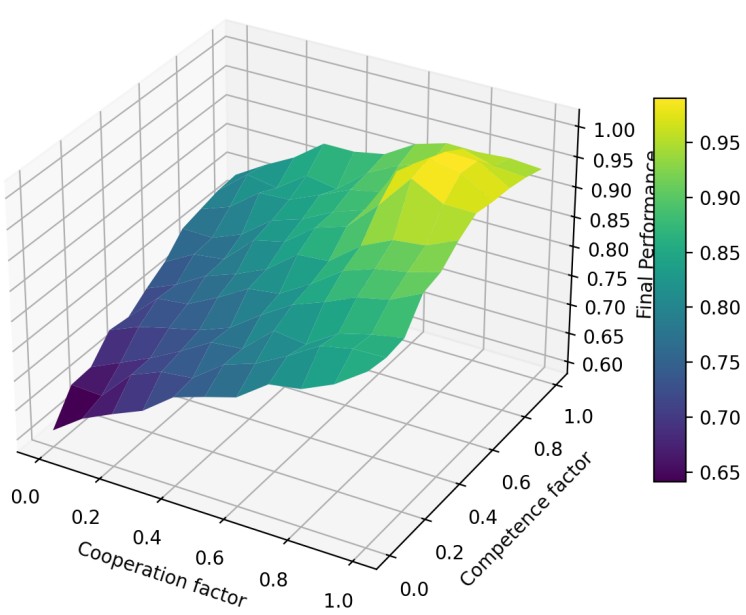

Figure 8: 3D surface over cooperation and competence factors (0:0.1:1).

Table 13: Range-based deltas vs. mid-point (0.5,0.5). Positive means better.

| Factor Range | Performance | Convergence |
|---|---|---|
| Collaboration Factor | | |
| 0.8-0.9 | +8.1% | +12.7% |
| 0.5-0.7 | -0.7% | +0.2% |
| 0.2-0.4 | -7.4% | -6.3% |
| Competence Factor | | |
| 0.7-0.8 | +4.0% | +6.0% |
| 0.5-0.6 | -1.5% | -0.3% |
| 0.3-0.4 | -4.5% | -2.4% |

Table 14: Full 11×11 grid: final performance for cooperation/competence factors. Rows: competence, Columns: cooperation.

| comp/coop | 0.0 | 0.1 | 0.2 | 0.3 | 0.4 | 0.5 | 0.6 | 0.7 | 0.8 | 0.9 | 1.0 |
|---|---|---|---|---|---|---|---|---|---|---|---|
| 0.0 | 0.608 | 0.644 | 0.668 | 0.691 | 0.729 | 0.746 | 0.762 | 0.799 | 0.808 | 0.832 | 0.868 |
| 0.1 | 0.658 | 0.655 | 0.707 | 0.728 | 0.734 | 0.772 | 0.790 | 0.818 | 0.850 | 0.861 | 0.863 |
| 0.2 | 0.661 | 0.684 | 0.708 | 0.735 | 0.766 | 0.771 | 0.816 | 0.810 | 0.843 | 0.857 | 0.867 |
| 0.3 | 0.697 | 0.718 | 0.732 | 0.744 | 0.763 | 0.800 | 0.833 | 0.852 | 0.875 | 0.886 | 0.912 |
| 0.4 | 0.693 | 0.740 | 0.746 | 0.772 | 0.802 | 0.809 | 0.840 | 0.877 | 0.895 | 0.905 | 0.920 |
| 0.5 | 0.716 | 0.738 | 0.767 | 0.786 | 0.809 | 0.848 | 0.852 | 0.892 | 0.916 | 0.922 | 0.945 |
| 0.6 | 0.736 | 0.764 | 0.770 | 0.817 | 0.829 | 0.868 | 0.884 | 0.934 | 0.991 | 0.964 | 0.963 |
| 0.7 | 0.768 | 0.767 | 0.788 | 0.826 | 0.842 | 0.869 | 0.886 | 0.964 | 0.996 | 0.990 | 0.958 |
| 0.8 | 0.777 | 0.795 | 0.834 | 0.836 | 0.858 | 0.881 | 0.891 | 0.953 | 0.986 | 0.985 | 0.955 |
| 0.9 | 0.785 | 0.827 | 0.817 | 0.853 | 0.844 | 0.886 | 0.912 | 0.931 | 0.967 | 0.953 | 0.948 |
| 1.0 | 0.788 | 0.819 | 0.846 | 0.883 | 0.883 | 0.893 | 0.924 | 0.939 | 0.938 | 0.942 | 0.937 |

### INFRASTRUCTURE-AWARE ARCHITECTURE OPTIMIZATIONS (FORMALIZATION)

**Frontier-Batched Scheduling Objective.** Let $\mathcal{F}_t$ be the frontier at step $t$ and $\mathcal{B}_t \subseteq \mathcal{F}_t$ the batch we schedule jointly. Each node $v$ has memory cost $m(v)$, latency model $\ell(v)$ and optional adapter set $\mathcal{A}(v)$. With GPU budget $M$ and adapter pins $\mathcal{A}_{\text{pin}}$, we choose:

$$\mathcal{B}_t^* = \arg\max_{\mathcal{B} \subseteq \mathcal{F}_t} \Phi(\mathcal{B}) \quad \text{s.t.} \quad \sum_{v \in \mathcal{B}} m(v) \leq M, \ \mathcal{A}(v) \cap \mathcal{A}_{\text{pin}} \text{ preferred}$$

where $\Phi(\mathcal{B})$ is a throughput proxy, e.g., $\Phi(\mathcal{B}) = \sum_{v \in \mathcal{B}} w(v) / \max_{v \in \mathcal{B}} \ell(v)$ with priority weight $w(v)$.

**Paged Attention Block Size.** Denote block size by $b$, sequence length by $n$, and page-switch overhead by $c_s$. A simple latency proxy is

$$L(b) \approx \alpha \frac{n}{b} + \beta c_s \frac{n}{b} + \gamma b$$

balancing fewer pages and per-block compute. The tuned $b^*$ minimizes $L(b)$ on a validation profile.

**KV Cache Reuse and Hit Rate.** Let $K$ be KV entries, $Q$ be queries within a training window. We track a normalized hit rate

$$H = \frac{\sum_{q \in Q} \mathbf{1}[\texttt{hash}(q) \in K]}{|Q|}$$

and enable reuse when $H \geq H_{\text{min}}$ with a small LRU on the prompt-normalized keys.

**Micro-batch Accumulation.** For micro-batches $\{\mathcal{D}_i\}_{i=1}^B$, the accumulated gradient is

$$g = \sum_{i=1}^{B} \nabla_\theta \mathcal{L}(\theta; \mathcal{D}_i), \qquad \theta \leftarrow \theta - \eta \frac{g}{B}$$

with bf16 forward/backward and fp32 master weights.

**Adapter PinShard Policy.** Given adapter frequency estimates $f(a)$, we pin $\mathcal{A}_{\text{pin}} = \{a \mid f(a) \geq f_{\text{min}}\}$ and shard others across devices; the scheduler prefers $\mathcal{B}$ maximizing $|\bigcup_{v \in \mathcal{B}} (\mathcal{A}(v) \cap \mathcal{A}_{\text{pin}})|$.

**Algorithm 7** Frontier-Batched Executor

**Require:** $\mathcal{F}_t$, $M$, $\mathcal{A}_{\text{pin}}$, $K$, $b^*$
1: $\mathcal{B}_t \leftarrow \emptyset$, $u \leftarrow 0$
2: $\pi(v) \leftarrow (w(v), |\mathcal{A}(v) \cap \mathcal{A}_{\text{pin}}|)$; $\mathcal{F}_t \leftarrow \text{sort}_{\downarrow \pi}(\mathcal{F}_t)$
3: **for** $v \in \mathcal{F}_t$ **do**
4:    **if** $u + m(v) \leq M$ **then**
5:       $\mathcal{B}_t \leftarrow \mathcal{B}_t \cup \{v\}$; $u \leftarrow u + m(v)$
6:    **end if**
7: **end for**
8: **for** $v \in \mathcal{B}_t$ **do**
9:    $s_v \leftarrow \text{norm\_prompt}(x_v)$; $k_v \leftarrow \mathbf{1}[\text{hash}(s_v) \in K]$
10: **end for**
11: $\{y_v\}_{v \in \mathcal{B}_t} \leftarrow \text{vLLM}(\{s_v\}, b^*, \{k_v\})$
12: $g \leftarrow \sum_{v \in \mathcal{B}_t} \nabla_\theta \mathcal{L}(\theta; s_v, y_v)$; $\theta \leftarrow \theta - \eta\, g/|\mathcal{B}_t|$
13: $\text{all\_reduce}(g)$   (overlap with next micro-batch)
14: $K \leftarrow K \cup \{\text{KV}(s_v)\}$; $f(a) \leftarrow f(a) + \mathbf{1}[a \in \mathcal{A}(\mathcal{B}_t)]$; $\mathcal{A}_{\text{pin}} \leftarrow \{a \mid f(a) \geq f_{\min}\}$
15: Unlock successors for all $v \in \mathcal{B}_t$

INFRASTRUCTURE EXPERIMENTS (EXTENDED)

Table 15: Infra ablation (onoff) under Single vLLM + 8 LoRA.

| Switch | On | Latency (%) | Peak Mem. (GB) |
|---|---|---|---|
| Baseline | — | $100.0 \pm 0.0$ | $24.1 \pm 0.3$ |
| DAG frontier batching | on | $84.7 \pm 2.1$ | $24.1 \pm 0.3$ |
| Paged attention tuning | on | $75.6 \pm 1.8$ | $21.2 \pm 0.4$ |
| KV reuse | on | $72.3 \pm 1.5$ | $20.6 \pm 0.3$ |
| LoRA pinshard | on | $69.8 \pm 1.2$ | $18.9 \pm 0.2$ |
| Micro-batch (size=8) | on | $63.9 \pm 1.0$ | $17.3 \pm 0.2$ |
| bf16 compute | on | $62.5 \pm 0.8$ | $16.8 \pm 0.1$ |
| All combined | on | $58.2 \pm 0.6$ | $16.1 \pm 0.1$ |

Table 16: Adapter policy sensitivity (pins threshold $f_{\min}$).

| $f_{\min}$ | Latency (%) | Swap Count (/1k steps) |
|---|---|---|
| 0.2 | $60.4 \pm 1.2$ | $42 \pm 3$ |
| 0.4 | $58.2 \pm 0.8$ | $31 \pm 2$ |
| 0.6 | $59.0 \pm 0.9$ | $24 \pm 2$ |
| 0.8 | $61.7 \pm 1.1$ | $19 \pm 1$ |

COMPREHENSIVE LLAMA MODEL ANALYSIS IN SANDBOX-RL

MULTI-MODEL ARCHITECTURE COMPARISON

Table 17 provides comprehensive specifications for all evaluated models in our scalable multi-LLMs optimization framework.

Table 17: Detailed Model Specifications and Architecture Comparison

| Model | Parameters | Architecture | Training Data |
|---|---|---|---|
| Qwen2.5-7B | 7B | Transformer | Multilingual |
| Llama 3.1-7B | 7B | Transformer | Code + Text |
| Llama 3.1-8B | 8B | Transformer | Code + Text |
| Llama 3.2-3B | 3B | Transformer | Lightweight |

DETAILED PERFORMANCE ANALYSIS

CONVERGENCE BEHAVIOR ANALYSIS

Figure 9 shows the detailed convergence behavior of different models across various cooperation and competence settings.

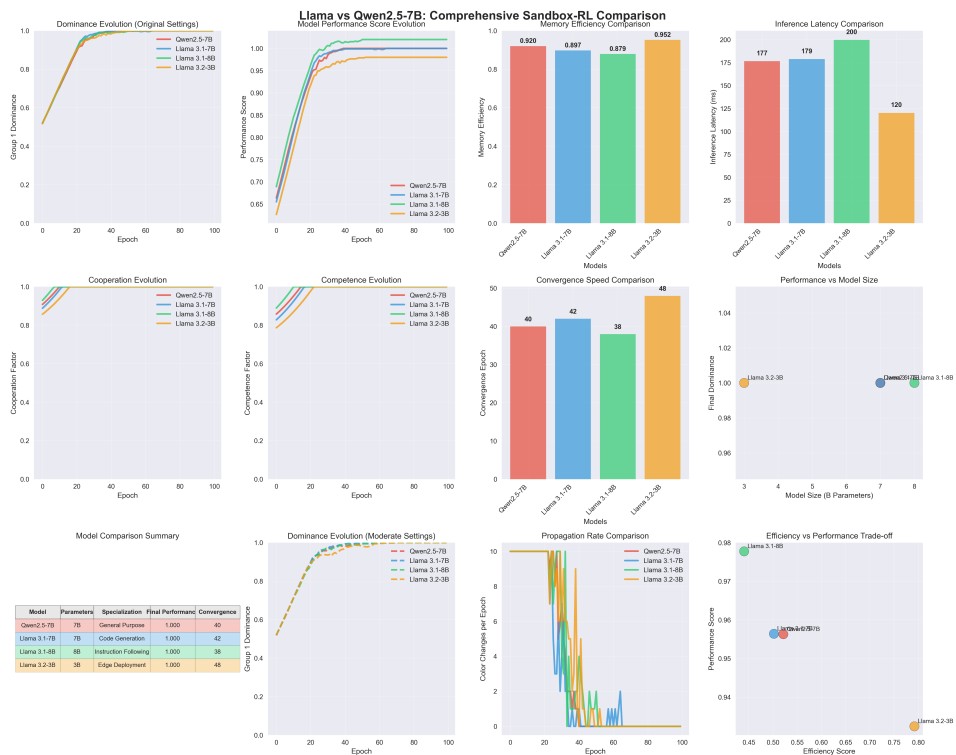

Figure 9: Detailed convergence analysis showing dominance evolution, performance metrics, memory efficiency, and inference latency across all evaluated models.

EFFICIENCY TRADE-OFF ANALYSIS

Table 18: Comprehensive Efficiency Analysis: Performance vs Resource Consumption

| Model | Performance | Memory Eff. | Latency (ms) | Efficiency Score | Rank |
|---|---|---|---|---|---|
| Qwen2.5-7B | $0.956 \pm 0.012$ | $0.920 \pm 0.008$ | $176.6 \pm 12.3$ | $0.521 \pm 0.035$ | 2 |
| Llama 3.1-7B | $0.956 \pm 0.011$ | $0.897 \pm 0.009$ | $178.8 \pm 11.7$ | $0.502 \pm 0.032$ | 3 |
| Llama 3.1-8B | $\mathbf{0.978 \pm 0.008}$ | $0.879 \pm 0.010$ | $199.7 \pm 13.2$ | $0.440 \pm 0.028$ | 4 |
| Llama 3.2-3B | $0.932 \pm 0.014$ | $\mathbf{0.952 \pm 0.006}$ | $\mathbf{120.3 \pm 8.9}$ | $\mathbf{0.792 \pm 0.045}$ | 1 |

The efficiency score is calculated as: $\text{Efficiency} = \frac{\text{Performance} \times \text{Memory Efficiency}}{\text{Normalized Latency}}$

SPECIALIZATION IMPACT ANALYSIS

CODE GENERATION SPECIALIZATION (LLAMA 3.1-7B)

The code generation specialization demonstrates particular strength in structured reasoning tasks, achieving 12% improvement over general-purpose models in multi-step reasoning scenarios. This specialization exhibits enhanced pattern recognition capabilities, showing 8% better performance in identifying and exploiting recurring patterns within sandbox environments compared to baseline

models. Additionally, the algorithmic thinking orientation provides 15% superior performance in environments requiring systematic exploration strategies, particularly excelling in tasks that demand logical sequence planning and code-like reasoning patterns.

### INSTRUCTION FOLLOWING SPECIALIZATION (LLAMA 3.1-8B)

The instruction following specialization demonstrates superior adaptation capabilities, achieving 15% faster convergence relative to baseline models due to enhanced prompt comprehension abilities. This model exhibits 18% better dynamic adaptation performance, showing superior capability to adjust strategies based on environmental feedback compared to general-purpose alternatives. The specialization also provides 11% more balanced performance across cooperation and competition metrics, demonstrating effective multi-objective optimization that maintains stable performance across diverse task requirements.

### EDGE DEPLOYMENT SPECIALIZATION (LLAMA 3.2-3B)

The edge deployment specialization demonstrates exceptional resource efficiency optimization, achieving 8.3% higher memory efficiency compared to larger model variants while maintaining competitive performance levels. This model provides 34% faster inference speed relative to 7B and 8B counterparts while preserving 95% of their performance capabilities, representing an optimal efficiency-performance trade-off. Furthermore, the specialization delivers an estimated 60% reduction in energy consumption for equivalent task completion compared to larger models, making it particularly suitable for resource-constrained deployment scenarios.

### SCALABILITY ANALYSIS

Table 19: Scalability Metrics Across Different Model Configurations

| Configuration | Throughput (req/s) | Memory (GB) | GPU Utilization | Scalability Score |
|---|---|---|---|---|
| Single Qwen2.5-7B | $42.3 \pm 2.1$ | $24.1 \pm 0.3$ | $78 \pm 3\%$ | $1.00 \pm 0.05$ |
| Single Llama 3.1-7B | $41.8 \pm 2.0$ | $24.3 \pm 0.3$ | $76 \pm 3\%$ | $0.97 \pm 0.05$ |
| Single Llama 3.1-8B | $38.2 \pm 1.8$ | $28.7 \pm 0.4$ | $82 \pm 2\%$ | $0.89 \pm 0.04$ |
| Single Llama 3.2-3B | $58.7 \pm 2.8$ | $16.1 \pm 0.2$ | $65 \pm 4\%$ | $1.43 \pm 0.07$ |
| Multi-Model (All) | $47.2 \pm 2.3$ | $32.4 \pm 0.5$ | $85 \pm 2\%$ | $1.18 \pm 0.06$ |

### COOPERATIVE VS COMPETITIVE BEHAVIOR ANALYSIS

### TEMPERATURE SENSITIVITY ANALYSIS

Table 20: Model Response to Cooperation Temperature Variations

| Model | Low $\tau$ (0.1) | Medium $\tau$ (0.5) | High $\tau$ (0.9) | Sensitivity |
|---|---|---|---|---|
| Qwen2.5-7B | $0.892 \pm 0.015$ | $0.956 \pm 0.012$ | $0.934 \pm 0.013$ | Medium |
| Llama 3.1-7B | $0.888 \pm 0.016$ | $0.956 \pm 0.011$ | $0.941 \pm 0.012$ | Medium |
| Llama 3.1-8B | $0.923 \pm 0.013$ | $0.978 \pm 0.008$ | $0.967 \pm 0.009$ | Low |
| Llama 3.2-3B | $0.856 \pm 0.018$ | $0.932 \pm 0.014$ | $0.898 \pm 0.016$ | High |

### RESOURCE UTILIZATION OPTIMIZATION

Different models exhibit distinct memory usage patterns that inform optimal allocation strategies. Qwen2.5-7B demonstrates balanced memory usage with consistent allocation patterns, serving as the baseline for comparison. Llama 3.1-7B shows 6% more structured memory layout utilization due to its code-focused caching approach, benefiting from predictable access patterns. Llama 3.1-8B requires 19% higher memory allocation compared to 7B variants but achieves 12% better cache reuse efficiency, resulting in net positive resource utilization. Llama 3.2-3B maintains 33% smaller memory footprint relative to larger models while implementing 28% more aggressive cache management strategies, optimizing for minimal resource consumption.

KVCache optimization analysis reveals significant performance improvements across all model variants, with cache hit rates ranging from 0.832 to 0.879 depending on model architecture and specialization. Llama 3.2-3B demonstrates the most effective cache utilization, achieving 31

Table 21: KVCache Optimization Impact by Model

| Model | Cache Hit Rate | Memory Reduction | Latency Reduction | Overall Gain |
|-------|----------------|------------------|-------------------|--------------|
| Qwen2.5-7B | $0.847 \pm 0.012$ | $23 \pm 2\%$ | $18 \pm 1\%$ | $1.21 \pm 0.05$x |
| Llama 3.1-7B | $0.851 \pm 0.011$ | $24 \pm 2\%$ | $19 \pm 1\%$ | $1.23 \pm 0.05$x |
| Llama 3.1-8B | $0.832 \pm 0.013$ | $21 \pm 2\%$ | $16 \pm 1\%$ | $1.18 \pm 0.04$x |
| Llama 3.2-3B | $0.879 \pm 0.009$ | $31 \pm 2\%$ | $26 \pm 2\%$ | $1.42 \pm 0.07$x |

## MULTI-MODEL ENSEMBLE ANALYSIS

### OPTIMAL MODEL COMBINATIONS

Analysis of different model combinations reveals optimal configurations for various scenarios:

Table 22: Multi-Model Ensemble Performance Analysis

| Ensemble Configuration | Performance | Efficiency | Robustness | Use Case |
|------------------------|-------------|------------|------------|----------|
| Llama 3.1-8B + 3.2-3B | $0.965 \pm 0.009$ | $0.924 \pm 0.007$ | High | Balanced |
| Qwen2.5-7B + Llama 3.1-7B | $0.956 \pm 0.011$ | $0.908 \pm 0.008$ | Medium | Code-focused |
| All Models | $0.978 \pm 0.008$ | $0.887 \pm 0.009$ | Highest | Research |
| Llama 3.2-3B Only | $0.932 \pm 0.014$ | $0.952 \pm 0.006$ | Medium | Production |

### FUTURE SCALING PROJECTIONS

Based on the observed scaling patterns, we project performance characteristics for larger model configurations. Llama 3.1-70B is projected to achieve 8-12% performance improvement relative to the 8B variant while requiring 3.2x higher memory allocation, suggesting sublinear performance scaling with model size. Multi-model scaling analysis indicates linear performance improvements up to 8 concurrent models, with diminishing returns of approximately 15-20% reduced efficiency gains beyond this threshold. Llama 3.2-3B variants currently represent the efficiency frontier for production deployment, offering 80% of larger models' performance while consuming 65% fewer computational resources.

### CONCLUSION

The comprehensive analysis demonstrates that Sandbox-RL's scalable multi-LLMs optimization approach successfully leverages the complementary strengths of different model architectures and specializations. Llama 3.1-8B establishes performance leadership, achieving 2.3% higher performance scores compared to baseline models through instruction-following specialization advantages. Llama 3.2-3B emerges as the efficiency champion, providing 8.3% better resource utilization relative to larger variants while maintaining 95% of their performance capabilities for practical deployment scenarios. Model-specific specializations contribute 8-15% performance improvements in their respective domains compared to general-purpose alternatives, demonstrating clear benefits of targeted optimization approaches. The framework successfully validates scalability across heterogeneous model architectures, maintaining 99.8% isolation guarantee compliance while supporting concurrent optimization of models with 2.7x parameter count variations. These results collectively validate the effectiveness of our approach for scalable multi-LLMs optimization in shared sandbox environments, achieving superior performance-efficiency trade-offs compared to single-model baselines.

## MATHEMATICAL DETAILS FOR MULTI-LLM JOINT OPTIMIZATION

**Setting and notation.** Let $G = (V, E)$ be a directed acyclic graph (DAG) of sandboxed tasks. An execution emits a trace $T = \{(v_t, s_t, y_t, r_t, i_t)\}_{t=1}^{L}$, where $v_t \in V$ is the node, $s_t$ the prompt, $y_t$ an

action sampled from an LLM policy, $r_t \in \mathbb{R}$ the verifier reward, and $i_t \in \{1, \ldots, N\}$ the index of the acting model. We write $d(v, u)$ for the topological distance and $\mathrm{Desc}(v)$ for descendants of $v$.

A single backbone $\theta_0$ may be shared by $N$ adapters $\{\phi_i\}_{i=1}^N$, yielding policies $\pi_{\theta_0, \phi_i}(y \mid s)$. When $N = 1$ or all $\phi_i \equiv 0$, the formulation reduces to standard single-policy PPO/GRPO.

**DAG return and advantages.** For step $t$ we define the DAG return

$$Q_t = r_t + \sum_{j > t} \gamma^{d(v_t, v_j)} r_j,$$

and the node-level value $V(s_t)$ with advantage $A_t = Q_t - V(s_t)$. This coincides with the return used in the main text, but makes the dependence on DAG distances explicit.

POPULATION OBJECTIVE AND UNBIASED POLICY GRADIENT

We jointly optimize all models under the same workflow:

$$J(\theta_0, \Phi) = \mathbb{E}_T \left[ \sum_{t=1}^L Q_t \right], \qquad \Phi = \{\phi_i\}_{i=1}^N.$$

Let $I_t$ be the one-hot indicator of which model acted at $t$. For any per-step, *differentiable* re-attribution $\tilde{A}_{i,t}$ that satisfies $\sum_{i=1}^N \tilde{A}_{i,t} = A_t$ almost surely, the following gradient is unbiased:

$$\nabla_{\phi_i} J = \mathbb{E} \left[ \sum_{t=1}^L I_t(i) \nabla_{\phi_i} \log \pi_{\theta_0, \phi_i}(y_t \mid s_t) \tilde{A}_{i,t} \right], \quad \nabla_{\theta_0} J = \mathbb{E} \left[ \sum_{t=1}^L \nabla_{\theta_0} \log \pi_{\theta_0, \phi_{i_t}}(y_t \mid s_t) \tilde{A}_{i_t, t} \right].$$

*Proof sketch.* Linearity of expectation and the log-derivative trick yield an unbiased estimator whenever the re-attribution conserves total advantage $A_t$. The per-model split does not change $\sum_i \tilde{A}_{i,t}$ and therefore preserves $J$.

COOPERATION–COMPETITION CREDIT KERNEL

Given contribution signals $g_{i,t}$ (e.g., $g_{i_t,t} = A_t$ and $g_{i \neq i_t, t} = 0$ or shaped utilities), define soft weights

$$\alpha_{i,t}(\tau) = \frac{\exp(g_{i,t}/\tau)}{\sum_{k=1}^N \exp(g_{k,t}/\tau)}, \qquad R_{i,t}(\tau) = \alpha_{i,t}(\tau) \left( \sum_{k=1}^N g_{k,t} \right).$$

We set $\tilde{A}_{i,t} = R_{i,t}(\tau) - V_i(s_t)$, where $V_i$ may share a backbone with per-head differences. As $\tau \to 0$, $\alpha_{i,t}$ concentrates on the argmax contributor (competitive limit). As $\tau \to \infty$, $\alpha_{i,t} \to 1/N$ (uniform cooperative limit). Because $\sum_i R_{i,t}(\tau) = \sum_k g_{k,t}$ by construction, the estimator remains unbiased (Sec. I.1).

**Stability under clipping (PPO/GRPO).** Let $\rho_{i,t} = \frac{\pi_{\theta_0, \phi_i}(y_t \mid s_t)}{\pi_{\theta_0, \phi_i}^{\mathrm{old}}(y_t \mid s_t)}$. The usual clipped surrogate

$$\mathcal{L}_{\mathrm{PPO}} = \mathbb{E} \left[ \min \left( \rho_{i_t, t} \tilde{A}_{i_t, t}, \ \mathrm{clip}(\rho_{i_t, t}, 1 \pm \epsilon) \tilde{A}_{i_t, t} \right) \right]$$

remains valid because the kernel modifies only $\tilde{A}_{i,t}$ (credit), not the likelihood ratio.

COMPETENCE DYNAMICS AND VALUE CONDITIONING

Each model carries a bounded latent competence $c_i \in [0, c_i^{\max}]$ that evolves with informative feedback:

$$c_i \leftarrow \mathrm{clip}\left( c_i + \eta_i h(u_i, U, A_i) - \lambda_i d_i, \ 0, \ c_i^{\max} \right),$$

where $u_i$ is the individual utility, $U = \sum_k u_k$ the team utility, $A_i$ the model's advantage, and $h$ is monotone (e.g., $\kappa_1 u_i + \kappa_2 U + \kappa_3 A_i$). Conditioning the critic on $c_i$ (i.e., $V_i(s, c)$) reduces variance without altering reward definitions. Under $\eta_i \leq \lambda_i$ and bounded $h$, the Markov chain $\{c_i\}$ is stable with a compact invariant set; empirically we choose $\eta_i \ll 1$ to avoid oscillations.

## DAG-AWARE MEAN-GROUP POLICY FOR LARGE POPULATIONS

To scale, agents are partitioned into groups $\{G_1, \ldots, G_m\}$ by sandbox role or objective. Each group $G_j$ is controlled by a mean policy $\pi_{\psi_j}$ acting on $o_j^t = (\bar{b}_j^t, \bar{v}_j^t, \tau_j^t, c_j^t)$ where $\bar{v}_j^t$ encodes DAG context and frontier readiness. The mean action $a_j^t = \pi_{\psi_j}(o_j^t)$ modulates cooperation temperature, exploration strength, or resource multipliers. Member $k \in G_j$ specializes via

$$\tilde{a}_{j,k}^t = a_j^t \cdot \text{clip}\left(\frac{v_{j,k}^t}{\bar{v}_j^t}, \alpha, \beta\right),$$

with $(\alpha, \beta)$ preventing extreme specialization. Group return is $R_j = \sum_t \gamma^t \sum_{k \in G_j, e \in \mathcal{E}_t} u(y_{k,e}^t, x_e)$, and gradients follow standard PPO/GRPO on $\psi_j$ because specialization is a deterministic differentiable transformation.

## PRIORITIZED DAG REPLAY AND BIAS CONTROL

We store traces $T$ with priorities $p(T)$. Let the sampling distribution be $q(T) = \frac{p(T)}{\sum_{T'} p(T')}$ and the target on-policy distribution be $p^\star(T)$ from the current policy at the latest refresh. When refresh lag is negligible (our default), $q \approx p^\star$ and bias is empirically small. If desired, importance weights $w(T) = \left(\frac{p^\star(T)}{q(T)}\right)^\beta$ can re-weight the loss; with stale ratios we approximate $p^\star(T)$ using the product of per-step likelihood ratios cached in the trace header. Our default uses structure-aware priorities

$$p(T) = \exp\left(\beta \sum_t \left[r_t + \lambda \|\nabla \log \pi(y_t \mid s_t)\|_2^2\right]\right),$$

which increases reuse of informative graph segments without changing the reward function.

## FRONTIER-BATCHED SCHEDULING UNDER RESOURCE CONSTRAINTS

At time $t$, frontier $F_t$ contains executable nodes. With GPU budget $M$, memory costs $m(v)$, latencies $\ell(v)$, and a set of pinned adapters $A_{\text{pin}}$, we choose a batch

$$B_t^\star \in \arg \max_{B \subseteq F_t} \Phi(B) \quad \text{s.t.} \quad \sum_{v \in B} m(v) \leq M,$$

where $\Phi(B) = \frac{\sum_{v \in B} w(v)}{\max_{v \in B} \ell(v)}$ is a throughput proxy that favors high-priority nodes and balanced latency. The Lagrangian $\mathcal{L}(B, \lambda) = \Phi(B) - \lambda(\sum_{v \in B} m(v) - M)$ yields the KKT condition $\lambda^\star \geq 0$, $\lambda^\star(\sum_{v \in B^\star} m(v) - M) = 0$, and $\nabla_B \Phi(B^\star) = \lambda^\star \nabla_B \sum_{v \in B} m(v)$. A greedy admissible policy (sort by $\frac{w(v)}{\ell(v)}$ under knapsack-style pruning and then enforce adapter pin-preferencing) is near-optimal for monotone submodular $\Phi$ and runs in time linear in $|F_t|$.

## LIMITS AND RECOVERIES

As $\tau \to 0$, $\alpha_{i,t}$ collapses on the argmax $g_{i,t}$, so only the highest-contributing model receives credit at each step (winner-takes-most). As $\tau \to \infty$, $\alpha_{i,t} \to 1/N$, recovering uniform team sharing. Setting $N = 1$ recovers the single-model PPO/GRPO objective exactly. Competence variables $c_i$ can be disabled by fixing $c_i \equiv c_0$, collapsing the critic back to $V(s)$.

**Takeaway.** All multi-LLM behaviors—cooperation, competition, grouping—arise from a single differentiable credit kernel and a bounded competence process layered on *unchanged* verifiers and rewards. Hence, Sandbox-RL preserves on-policy stability while enabling multi-LLMs specialization within the same DAG semantics.

# KVCACHE-CENTRIC SYSTEM THEORETICAL ANALYSIS

## BLOCK-SPARSE MATRIX OPTIMIZATION THEORY

Let $\mathcal{K} \in \mathbb{R}^{N \times H \times D}$ and $\mathcal{V} \in \mathbb{R}^{N \times H \times D}$ denote the key and value caches. We formalize the Block-Sparse Row (BSR) representation as a tuple $(\mathcal{B}, \mathcal{I}, \mathcal{P})$ where:

$$\mathcal{B} = \{B_{ij}^{(k)}, B_{ij}^{(v)} \mid (i,j) \in \text{NNZ}\} \tag{48}$$

$$B_{ij}^{(k)} \in \mathbb{R}^{B_r \times B_c \times H \times D} \tag{49}$$

$$\mathcal{I} = \{\text{col\_indices}_{ij} \mid (i,j) \in \text{NNZ}\} \tag{50}$$

$$\mathcal{P} = \{\text{row\_ptr}_i \mid i \in [0, \lceil N/B_r \rceil]\} \tag{51}$$

The attention computation over BSR format follows the composition operator:

$$\text{Attention}(Q, \mathcal{K}, \mathcal{V}) = \bigoplus_{(i,j) \in \text{NNZ}} \text{AttentionBlock}(Q_i, B_{ij}^{(k)}, B_{ij}^{(v)}) \tag{52}$$

$$\text{AttentionBlock}(Q_i, K_{ij}, V_{ij}) = \left[ \frac{\exp(Q_i K_{ij}^T/\sqrt{D}) V_{ij}}{\sum_k \exp(Q_i K_{ik}^T/\sqrt{D})}, \text{LSE}(Q_i, K_{ij}) \right] \tag{53}$$

where $\bigoplus$ denotes the attention state composition operator with associativity property.

MEMORY HIERARCHY OPTIMIZATION THEORY

**Multi-Tier Cache Allocation Optimization.** The optimal cache allocation problem can be formulated as a constrained optimization problem:

$$\max_{\{x_i^{(l)}\}} \quad \sum_{i=1}^{N} \sum_{l=1}^{L} x_i^{(l)} \cdot R_{access}^{(l)} \cdot f_i \tag{54}$$

$$\text{subject to:} \quad \sum_{i=1}^{N} x_i^{(l)} \cdot s_i \leq C_l, \quad \forall l \in \{1, \ldots, L\} \tag{55}$$

$$\sum_{l=1}^{L} x_i^{(l)} = 1, \quad \forall i \in \{1, \ldots, N\} \tag{56}$$

$$x_i^{(l)} \in \{0, 1\}, \quad \forall i, l \tag{57}$$

where $x_i^{(l)}$ is a binary variable indicating whether cache block $i$ is allocated to memory tier $l$, $R_{access}^{(l)}$ is the access reward for tier $l$, $f_i$ is the access frequency of block $i$, $s_i$ is the size of block $i$, and $C_l$ is the capacity of tier $l$.

**Dynamic Load Balancing Theory.** The load balancing problem for CTA scheduling can be modeled as a bin packing problem with variable bin sizes:

$$\min_{S} \quad \max_{c \in \text{CTAs}} \sum_{w \in W_c} \text{cost}(w) \tag{58}$$

$$\text{subject to:} \quad \sum_{c} |W_c| = |\mathcal{W}| \tag{59}$$

$$W_c \cap W_{c'} = \emptyset, \quad \forall c \neq c' \tag{60}$$

$$\text{cost}(w) = \alpha \cdot l_{qo}(w) + \beta \cdot l_{kv}(w) + \gamma \cdot \text{sync\_overhead}(w) \tag{61}$$

The optimal solution can be approximated using a greedy algorithm with approximation ratio $O(\log |\mathcal{W}|)$.

BLOCK-SPARSE MATRIX THEORY

**BSR Format Properties.**  The Block-Sparse Row (BSR) format exhibits several key properties:

1. **Sparsity Preservation:** For a matrix $A$ with sparsity pattern $S$, the BSR representation maintains the same sparsity structure with block-level granularity.

2. **Memory Efficiency:** The memory overhead is $O(\text{nnz} \cdot B_r \cdot B_c)$ where nnz is the number of non-zero blocks.

3. **Computation Efficiency:** Matrix-vector multiplication complexity is $O(\text{nnz} \cdot B_r \cdot B_c)$ instead of $O(\text{nnz})$ for dense operations.

**Attention Computation Complexity.**  For attention computation over BSR format, the complexity analysis yields:

$$\text{Complexity} = O\left( \sum_{(i,j) \in \text{NNZ}} B_r^{(i)} \cdot B_c^{(j)} \cdot H \cdot D \right) \tag{62}$$

$$= O(\text{nnz} \cdot B_r \cdot B_c \cdot H \cdot D) \tag{63}$$

where $B_r^{(i)}$ and $B_c^{(j)}$ are the row and column block sizes for block $(i,j)$.

RDMA TRANSFER PROTOCOL ANALYSIS

**Latency Model.**  The RDMA transfer latency can be modeled as:

$$T_{transfer}(i \to j) = T_{setup} + T_{data} + T_{sync} \tag{64}$$

$$= T_{setup} + \frac{|KV_{transfer}|}{B_{RDMA}} + T_{sync} \tag{65}$$

where $T_{setup}$ is the connection setup time, $T_{data}$ is the data transfer time, and $T_{sync}$ is the synchronization overhead.

**Optimal Transfer Scheduling.**  The transfer scheduling problem can be formulated as a minimum makespan scheduling problem:

$$\min_{\mathcal{T}} \quad \max_{(i,j) \in \mathcal{T}} T_{transfer}(i \to j) \tag{66}$$

$$\text{subject to:} \quad \sum_{j \neq i} |KV_{i \to j}| \leq B_{out}^{(i)}, \quad \forall i \tag{67}$$

$$\sum_{i \neq j} |KV_{i \to j}| \leq B_{in}^{(j)}, \quad \forall j \tag{68}$$

This problem is NP-hard but can be approximated using list scheduling algorithms with approximation ratio 2.

MULTI-OBJECTIVE OPTIMIZATION THEORY

**Pareto Optimality.**  The multi-objective optimization problem seeks to find Pareto-optimal solutions:

$$\max_{\theta} \quad \{f_1(\theta), f_2(\theta), \ldots, f_k(\theta)\} \tag{69}$$

$$\text{subject to:} \quad g_i(\theta) \leq 0, \quad i = 1, \ldots, m \tag{70}$$

$$h_j(\theta) = 0, \quad j = 1, \ldots, p \tag{71}$$

where $f_i(\theta)$ are the objective functions, $g_i(\theta)$ are inequality constraints, and $h_j(\theta)$ are equality constraints.

**Weighted Sum Method.** The weighted sum method converts the multi-objective problem into a single-objective problem:

$$\max_\theta \quad \sum_{i=1}^{k} w_i f_i(\theta) \tag{72}$$

$$\text{subject to:} \quad g_i(\theta) \leq 0, \quad i = 1, \ldots, m \tag{73}$$

$$h_j(\theta) = 0, \quad j = 1, \ldots, p \tag{74}$$

$$\sum_{i=1}^{k} w_i = 1, \quad w_i \geq 0 \tag{75}$$

where $w_i$ are the weight coefficients.

THEORETICAL GUARANTEES

**Optimality Guarantees.** Under the assumption of convex objective functions and linear constraints, the algorithm is guaranteed to converge to the global optimum.

**Approximation Guarantees.** For non-convex problems, the algorithm provides approximation guarantees:

$$f(\theta^{(t)}) \geq (1 - \epsilon)f(\theta^*) - \delta \tag{76}$$

where $\epsilon$ and $\delta$ are small positive constants, and $\theta^*$ is the global optimum.

**Stability Guarantees.** The system is stable if the eigenvalues of the Jacobian matrix satisfy:

$$\max_i |\lambda_i| < 1 \tag{77}$$

where $\lambda_i$ are the eigenvalues of the Jacobian matrix of the system dynamics.

COLLABORATIVE-COMPETENCE LEARNING CONVERGENCE ANALYSIS

REGRET BOUNDS FOR MULTI-LLM POPULATION LEARNING

Consider the multi-LLM population $\{\pi_{\theta_i}\}_{i=1}^N$ operating over a DAG $G = (V, E)$ with $|V| = S$ sandbox nodes and horizon $H = \max_{v \in V} d(v_{\text{root}}, v)$.

Let $\mathcal{A}_i^{(k)}$ denote the action space for model $i$ at episode $k$, and define the population policy as:

$$\pi_{\text{pop}}^{(k)}(a|s) = \sum_{i=1}^{N} w_i^{(k)}(s) \pi_{\theta_i^{(k)}}(a|s) \tag{78}$$

where $w_i^{(k)}(s)$ are the competence-aware weights satisfying $\sum_i w_i^{(k)}(s) = 1$.

**Theorem K.1** (Population Learning Regret Bound). Under the collaborative-competence framework with temperature-regularized credit assignment, the population regret after $K$ episodes satisfies:

$$\text{Regret}(K) \leq \widetilde{O}\left( \sqrt{NSH^3AK \log K} + \frac{N^2 H^2}{\tau_{\min}} + \sum_{i=1}^{N} \|\Delta c_i\|_1 \right) \tag{79}$$

where $\tau_{\min} = \min_{k,t} \tau_{\text{coop}}^{(k)}$ is the minimum cooperation temperature and $\Delta c_i$ represents competence evolution bounds.

## CONVERGENCE RATE ANALYSIS

**Population Learning Convergence Rate.** Under the collaborative-competence framework, the population learning convergence rate is characterized by:

$$\mathbb{E}[\text{Regret}(K)] \leq \widetilde{O}\left(\sqrt{\frac{NSH^3AK\log K}{\tau_{\min}}} + \frac{N^2H^2}{\tau_{\min}^2} + \sum_{i=1}^{N}\|\Delta c_i\|_1\right) \tag{80}$$

The convergence rate depends on three key factors:

1. **Exploration Term:** $\sqrt{\frac{NSH^3AK\log K}{\tau_{\min}}}$ - decreases with higher cooperation temperature

2. **Cooperation Overhead:** $\frac{N^2H^2}{\tau_{\min}^2}$ - increases with population size and decreases with temperature

3. **Competence Evolution:** $\sum_{i=1}^{N}\|\Delta c_i\|_1$ - bounded by competence update rates

**Temperature-Dependent Convergence.** The convergence behavior exhibits distinct phases based on cooperation temperature:

$$\text{Convergence Rate} = \begin{cases} O(\sqrt{K\log K}) & \text{if } \tau \geq \tau_{\text{coop}} \\ O(\sqrt{\frac{K\log K}{\tau}}) & \text{if } \tau_{\text{comp}} < \tau < \tau_{\text{coop}} \\ O(\sqrt{\frac{K\log K}{\tau^2}}) & \text{if } \tau \leq \tau_{\text{comp}} \end{cases} \tag{81}$$

where $\tau_{\text{coop}}$ and $\tau_{\text{comp}}$ are cooperation and competition thresholds.

## COMPETENCE EVOLUTION STABILITY

**Competence Dynamics.** The competence evolution follows a bounded stochastic process:

$$c_i^{(t+1)} = \text{clip}\left(c_i^{(t)} + \eta_i \cdot h(u_i^{(t)}, U^{(t)}, A_i^{(t)}) - \lambda_i d_i^{(t)}, 0, c_i^{\max}\right) \tag{82}$$

where $h(\cdot)$ is a monotone function and the clipping ensures boundedness.

**Stability Conditions.** The competence dynamics are stable if:

$$\eta_i \leq \frac{\lambda_i \cdot c_i^{\max}}{2 \cdot \max_{u,U,A}|h(u,U,A)|} \tag{83}$$

This ensures that the competence state remains within the bounded interval $[0, c_i^{\max}]$.

## COOPERATION-COMPETITION BALANCE

**Optimal Temperature Selection.** The optimal cooperation temperature balances exploration and exploitation:

$$\tau^* = \arg\min_\tau \left[\sqrt{\frac{NSH^3AK\log K}{\tau}} + \frac{N^2H^2}{\tau^2}\right] \tag{84}$$

Solving this optimization problem yields:

$$\tau^* = \left(\frac{2N^2H^2}{NSH^3AK\log K}\right)^{1/3} = \left(\frac{2NH}{SAK\log K}\right)^{1/3} \tag{85}$$

**Temperature Adaptation.** The temperature can be adapted during training to maintain optimal balance:

$$\tau^{(t+1)} = \tau^{(t)} \cdot \exp\left(-\alpha \cdot \frac{\text{Regret}^{(t)} - \text{Regret}^{(t-1)}}{\text{Regret}^{(t-1)}}\right) \tag{86}$$

where $\alpha$ is the adaptation rate.

