# OpenReview forum: "Sandbox-RL: Scalable Multi-LLMs Optimization through Sandbox-Based Reinforcement Learning"
_ICLR.cc/2026/Conference — ICLR 2026 Conference Withdrawn Submission_

### Official Review · Reviewer_U95w · 2025-10-14

**Soundness:** 3
**Presentation:** 2
**Contribution:** 2
**Rating:** 4
**Confidence:** 3

**Summary:**

This paper proposes Sandbox-RL, a reinforcement learning framework for optimizing multiple large language models through structured sandbox environments organized as workflow graphs. The method enables heterogeneous models to co-train under controlled cooperation and competition, supported by a scalable system architecture that improves reward attribution and training efficiency. Experimental results show faster convergence and better performance than existing multi-agent or single-model RL approaches.

**Strengths:**

1. Proposes a novel framework for multi-LLM reinforcement learning, integrating heterogeneous models through structured DAG-based sandbox environments.
2. Introduces principled temperature-regularized cooperation and competence-aware specialization mechanisms that unify cooperative and competitive dynamics.
3. Demonstrates strong and consistent empirical gains across multiple LLM families and reasoning tasks, supported by detailed ablations and system-level optimizations.

**Weaknesses:**

1. Limited validation scope: Although the experiments cover multiple reasoning and simulation benchmarks, all evaluations remain text-based. The framework’s claimed scalability and generality (e.g., “multi-LLM co-optimization across heterogeneous tasks”) would be more convincing if validated in open-ended or real-world multi-agent environments such as tool-use or embodied reasoning.
2. Lack of clear causal attribution for improvements: The paper reports large performance gains (e.g., +8.7% to +101%) but does not disentangle how much comes from the proposed cooperation–competence mechanism versus from system-level KVCache optimizations or architectural parallelization. A controlled ablation isolating algorithmic vs. infrastructural effects is needed.
3. Theoretical and empirical depth mismatch: While the framework introduces formal definitions and claims unbiased policy gradients, the derivations in the appendix are descriptive rather than rigorous, and no formal convergence or variance analysis is experimentally validated. Strengthening the theoretical grounding or providing empirical verification (e.g., variance reduction or stability plots) would improve credibility.

**Questions:**

Presentation suggestions
1. Move Figure 2 (system overview) to the beginning as a pull figure to help readers grasp the overall framework early.
2. Fix Figure 1: the legend overlaps with text and some side labels are too small to read.
3. Enlarge all text and axis labels in Figure 3, which are currently too small for legibility in the printed format.

---

### Official Review · Reviewer_b8J5 · 2025-11-01

**Soundness:** 2
**Presentation:** 1
**Contribution:** 2
**Rating:** 2
**Confidence:** 4

**Summary:**

This paper presents Sandbox-RL, a system-level framework for multi-LLM reinforcement learning, aiming to jointly optimize heterogeneous models (Qwen, Llama families) within modular sandbox environments structured as workflow DAGs. Each sandbox defines generation, prompting, and verification modules with isolated reward channels, enabling reproducible supervision and parallelized task composition. Reported results show up to +101% improvement in trading simulation and 14–35% gains in math reasoning accuracy, with 3–4× faster convergence and 40% lower memory use.

**Strengths:**

1. The topic and the goal of this paper is very interesting and important for the LLM RL field.
2. The sandbox-based workflow graph is a neat abstraction enabling modular evaluation and reproducible reward attribution across tasks.
3. KVCache-centric optimization and distributed scheduling demonstrate practical awareness of training constraints and scalability challenges

**Weaknesses:**

1. The presentation of this paper is not good enough. Figures and tables are visually cluttered and poorly organized: font sizes in captions and axis labels are too small to read, some captions overlap with figure boundaries, and color choices. The tables are not well-polished as well. The formatting issues significantly hurt readability and give the impression of a rushed, unpolished submission.
2. Baselines like “Always Cooperate/Compete” are toy setups and fail to include stronger LLM-RL frameworks (e.g., AgentGym-RL)
3. Reported metrics (e.g., 0.982 vs 0.903) lack standard deviations or confidence intervals.
4. Although system throughput and memory are discussed, there is no correlation analysis between system optimization and learning efficiency—unclear whether faster convergence stems from RL design or better scheduling.

**Questions:**

1. Can you provide full quantitative drops when removing each component (temperature regulation, competence states, KVCache optimization)?
2. What are compute costs (GPU-hours, memory) per model? How does the framework scale to >10 LLMs?
3. Will sandbox task definitions and reward verifiers be released to allow independent replication?

---

### Official Review · Reviewer_njSa · 2025-11-01

**Soundness:** 2
**Presentation:** 2
**Contribution:** 2
**Rating:** 2
**Confidence:** 4

**Summary:**

This paper introduces Sandbox-RL, a framework designed for the scalable co-optimization of multiple, heterogeneous LLMs. The core idea is to move away from traditional multi-agent systems that rely on inter-agent communication. Instead, Sandbox-RL orchestrates a population of LLMs (e.g., Qwen and Llama variants) within "structured workflow graphs" composed of modular "sandbox environments". Each sandbox is an isolated module with its own case generator, prompt function, and scoring mechanism, which the authors claim enables precise reward attribution and reusable learning signals. The framework manages the multi-LLM population through temperature-regularized optimization, using "competence matrices" and a "cooperation temperature" parameter to control the balance between competitive and cooperative behaviors. The system is supported by a KVCache-centric architecture, featuring distributed memory and intelligent scheduling to enhance efficiency. The authors evaluate Sandbox-RL on tasks including misinformation propagation (OASIS), a trading simulation, and math reasoning (GSM8K, MATH), reporting superior performance and efficiency trade-offs compared to several baseline methods.

**Strengths:**

The paper implements a complex system with many optimizations. The multi-LLM training is a timely problem.

**Weaknesses:**

- The paper suffers from a fundamental ambiguity regarding its primary contribution. It is unclear whether Sandbox-RL is intended as a novel *algorithm* or as a specialized *infrastructure* (system) for multi-LLM RL. It is described as a "framework" and "system", yet it also introduces specific algorithmic mechanisms like temperature-regularized credit and competence states. This system appears to be heavily constrained to a specific on-policy, PPO-style algorithm, and it is not demonstrated how it would support other common RL paradigms (e.g., offline RL, off-policy algorithms, or even SFT). If the contribution is algorithmic, the techniques themselves are not especially novel and resemble existing concepts in population-based training and MARL. This ambiguity weakens the paper's "principled" and "general" claims.

- The claims of "scalability" are not convincingly supported by the experiments. The LLMs used (3B, 7B, 8B parameters) are not considered "large-scale" by contemporary standards. While scalability might be interpreted as the *number* of agents, the experiments presented are also small-scale (e.g., 8 LoRA adapters for OASIS, 4-6 agents for other tasks). This evaluation does not substantiate the claims of a highly scalable system, such as the one scaling to "1000+ models" mentioned in Appendix C.

- There is a significant mismatch between the paper's stated motivation and its experimental validation. The introduction explicitly cites "software engineering" as a key example of a multi-actor task that would benefit from this framework. However, the experiments are limited to synthetic simulations (OASIS, Trading) and math benchmarks. These tasks do not reflect the complexity, long-horizon dependencies, or vast state spaces of the motivating software engineering example, making it difficult to assess if the framework would generalize to such challenging, real-world problems.

- The concept of "structured tasks" or "structured workflow execution" is foundational to the paper but is never clearly defined. It is not clear what properties a task must possess to be compatible with the DAG-based sandbox formalism, or what makes this approach fundamentally superior to standard MDP/POMDP formulations.

- The overall presentation of the paper needs improvement. The writing is often dense, and the citation format used in the bibliography is inconsistent and should be corrected.

**Questions:**

- In the sandbox environment formalism (equations 1-4), the interaction loop (`case_generator`, `prompt_func`, `pi_theta(s_i)`, `verify_score`) appears to be a single-turn, atomic operation. How does this formalism support multi-turn interactions or long-horizon decision-making *within* a single sandbox environment? Or is the long-horizon aspect handled *only* by the DAG structure, where each node is strictly a single-shot task?

- How is the workflow graph structure $\mathcal{G}=(V,E)$ determined for a given task? Is this graph manually specified by the user? If so, this would require significant, task-specific domain knowledge and would seem to be a major limitation on the framework's generality and practical applicability.

- How is the cooperation temperature parameter $\tau$ set? Is it a fixed hyperparameter that must be tuned, or is it adapted dynamically during training? The paper discusses its *effects* extensively but not its selection or optimization.

- The experimental comparison is limited to simple baselines like standard Policy Gradient (PG), Always Cooperate (AC), and Always Compete (AP). Why was Sandbox-RL not compared against established multi-agent RL (MARL) algorithms, even those that rely on communication? Such a comparison would be crucial for understanding the true benefits of the proposed sandbox-based isolation approach versus state-of-the-art MARL.

---

### Note · Authors · 2025-11-12

I have read and agree with the venue's withdrawal policy on behalf of myself and my co-authors.